# Study on Three-Dimensional Stress Field of Gob-Side Entry Retaining by Roof Cutting without Pillar under Near-Group Coal Seam Mining

**Xiaoming Sun** [1,2,*,†], **Yangyang Liu** [1,2,*,†], **Junwei Wang** [1,2], **Jiangbing Li** [1,2], **Shijie Sun** [1,2] **and Xuebin Cui** [1,2]

1    State Key Laboratory for Geomechanics & Deep Underground Engineering, China University of Mining & Technology, Beijing 100083, China
2    School of Mechanics and Civil Engineering, China University of Mining & Technology, Beijing 100083, China
*    Correspondence: zqt1700620120g@student.cumtb.edu.cn (X.S.);
     zqt1700620112g@student.cumtb.edu.cn (Y.L.); Tel.: +86-180-3922-5611 (X.S.)
†    These authors contributed equally to this work.

**Abstract:** In order to explore the distribution law of stress field under the mining mode of gob-side entry retaining by roof cutting without pillar (GERRCP) under goaf, based on the engineering background of 8102 and 9101 working faces in Xiashanmao coal mine, the stress field distribution of GERRCP and traditional remaining pillar was studied by means of theoretical analysis and numerical simulation. The simulation results showed that: (1) in the front of the working face, the vertical peak stress of non-pillar mining was smaller than that of the remaining pillar mining, and it could effectively control stress concentration in surrounding rock of the mining roadway; the trend of horizontal stress distribution of the two was the same, and the area, span and peak stress of stress the rise zone were the largest in large pillar mining and the minimum in non-pillar mining. (2) On the left side of the working face, the vertical stress presented increasing-decreasing characteristics under non-pillar mining mode and saddle-shaped distribution characteristics under the remaining pillar mining mode respectively. Among them, the peak stress was the smallest under non-pillar mining, and compared with the mining of the large pillar and small pillar, non-pillar mining decreased by 12–21% and 3–10% respectively. The position of peak stress of the former was closer to the mining roadway, indicating that the width of the plastic zone of the surrounding rock of the non-pillar mining was smaller and bearing capacity was higher. In the mining of the large and small pillar, the horizontal stress formed a high stress concentration in the pillar and 9102 working face respectively. In non-pillar mining, the horizontal stress concentration appeared in solid coal, but the concentration area was small.

**Keywords:** non-pillar; gob-side entry retaining by roof cutting; close distance coal seams; goaf; stress distribution

---

## 1. Introduction

In the 1960s and 1970s, longwall mining technology developed rapidly, and the "masonry beam theory" was put forward, forming the "121" construction method of longwall mining [1]. This technology requires two roadways to be tunneled for each working face, and a large pillar is set up to balance underground pressure. The "transfer rock beam theory" was put forward by analyzing the existence of the internal and external stress field in a high-stress area, forming the "121" small pillar construction method of longwall mining [2,3]. However, the traditional mining method of "121" will form a hanging roof with insufficient collapse at the side of the goaf, and the roof subsidence and

rotary deformation are large, which greatly affects the stability of the pillar and support system on roadway side. In order to reduce the development ratio, increase the coal mining rate and improve the periodic pressure of roof, the "cutting cantilever beam theory" was born [4], and based on this theory, the mining technology of gob-side entry retaining by roof cutting without pillar (GERRCP) was proposed. In the new mining technology of GERRCP, only one roadway needs to be tunneled for each working face, and the other roadway is formed automatically by roof cutting and pressure relief. Moreover, there is no need to leave pillars, which reduces the waste of coal resources and avoids roof accidents, rock burst, gas outburst and other safety hazards caused by remaining pillars [5]. The phenomenon of stress concentration caused by a pillar is eliminated, and the pressure distribution of a stope is optimized, which makes coal mining more safe and efficient. Many scholars have carried out a lot of related research work on GERRCP using theoretical analysis, numerical simulation, laboratory experiments and field experiments. As one of the powerful methods, the numerical simulation has the advantage of low cost, high efficiency and good repeatability. It has been widely used in the related research of this technology and achieved good application results. With the introduction of GERRCP, its design principle and key technologies have been extensively investigated [6–8]. Guo et al. [9] studied the relationship between roof fracturing angle and stability of gob-side entry subjected to dynamic loading through establishing a numerical calculation model. Zhen et al. [10] investigated the influence of two methods of non-pillar-mining techniques by roof cutting and by filling artificial materials on the results of the entry retained via industrial case and numerical simulation. Guo et al. [11] studied the roof pre-fracturing and energy-absorbing support systems to evaluate the stress distribution and deformation control of gob-side entry by numerical simulation. Hu et al. [12] investigated the key parameters affecting GERRCP by theoretical analysis and numerical simulation. Combined with the above research, it can be found that the above researches on GERRCP were carried out under the condition of single coal-seam mining, and few researches on this technology when mining close distance coal seam. Therefore, it is of great significance to carry out relevant researches on GERRCP under the condition of near-group coal-seam mining.

The near-group coal-seam mining is very characteristic. When mining close distance coal-seam, the roof caving of the upper coal seam will cause various degrees of damage to the roof of the lower coal seam. As a result, the upper overburden structure and temporal and spatial distribution characteristics of the stress field during the mining of the lower coal seam are significantly different from those of a single coal seam. In particular, the mining direction of the lower coal seam is perpendicular to that of the upper coal seam, forming the vertical cross mining. Therefore, in order to ensure the safety of coal mining, it is of significance to analyze the distribution law of the stress field in lower coal seam when mining close distance coal seam. At present, domestic and foreign experts and scholars have conducted a lot of studies on the distribution law of a stress field in lower coal seam when mining close distance coal seam, and achieved fruitful results. Singh [13] established a numerical model and combined it with a double-yield model to assess its effectiveness in simulating the recovery of stress in goaf. Through theoretical analyses and physical modelling studies, the interaction between vertical stress distribution within goaf and surrounding rock mass in these systems was studied [14]. Zhang et al. [15] investigated the stress distribution, fracture development, and strata movement above a protective coal seam in longwall mining through numerical calculation. Liu et al. [16] analysed the stress distribution and roadway position of lower seams in the close distance coal seams by using numerical simulation. Zhang et al. [17] studied the floor failure depth of upper coal seam during close coal seams mining by building the mechanical model of floor failure of upper coal seam. Xu et al. [18] studied the stress propagation and distribution of a roadway by Kirsch equations and analyzed the changes of stress, displacement, and plastic zones around roadways during the mining of the upper coal seams by means of numerical simulation. Wang et al. [19] analyzed some key issues about abutment pressure and stress concentration shell by numerical simulations to study the distribution and evolution characteristics of the macroscopic stress field of surrounding rocks. Ma et al. [20] studied the stress distribution and deformation law of surrounding rocks for the water-dripping roadway below a contiguous seam goaf.

In-depth studies on the movement and instability characteristics and mining stress evolution law of the secondary mining structure of roof under goaf, were carried out by means of theoretical analysis, similarity simulation experiment, numerical simulation and field measurement [21–24].

The above experts and scholars have achieved fruitful results in research on the distribution law of a stress field when mining near-group coal seam. However, most previous studies have focused on conventional mining methods; few scholars have carried out relevant research on the stress distribution of the stope in GERRCP in the near-group coal-seam mining. In order to explore the distribution law of three-dimensional stress field of GERRCP in the near-group coal seam mining, this paper takes Xiashanmao coal mine as an engineering background, establishes the mechanical model of the roof structure of GERRCP through theoretical analysis, and establishes a three-dimensional numerical calculation model based on the finite difference program FLAC-3D (ITASCA, US) to study the distribution law of stress in the stope. Finally, the numerical simulation results are validated by field experiments.

## 2. Gob-Side Entry Retaining by Roof Cutting without Pillar (GERRCP)

### 2.1. Principle of GERRCP

As shown in Figure 1, the GERRCP adopts energy-gathered blasting technology to carry out advanced pre-splitting on the roof. The roof is cut off along the pre-splitting damaged structural surface through periodic weighting of the stope, and the fractured roof collapses naturally with the help of underground pressure. The side of the roadway is formed by the broken expanded characteristic of the collapsed gangue, and the flexible support body in a roadway is formed by the sliding and yielding gangue support structure and constant pressure retractable support equipment, which separates the goaf. At the same time, the high strength support of a roadway roof is formed by the constant-resistance large-deformation anchor cable (CRLDAC) with structural characteristics of negative Poisson's ratio, thus realizing the non-pillar mining of a single working face and single roadway [25,26]. This technology realizes the transformation of a roadway roof structure from a long-wall beam to short-wall beam by roof cutting, which ensures the stability of the roadway, can weaken the concentrated stress on the upper part of the coal body adjacent to working face, and can also avoid the roof collapse, rock burst and gas outburst caused by remaining pillars.

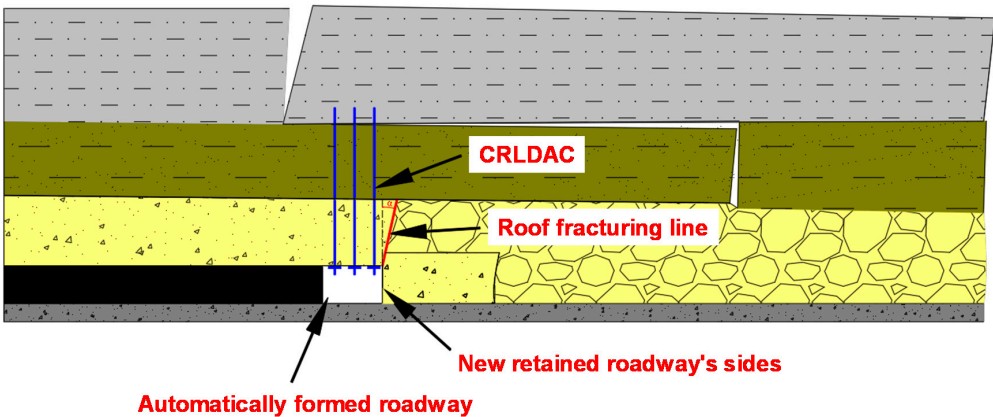

**Figure 1.** Section diagram of roof structure of gob-side entry retaining by roof cutting without pillar (GERRCP).

The mechanical model of the roof structure of GERRCP is introduced below (as shown in Figure 2). Under the action of periodic pressure, the immediate roof and main roof fracture and rotate. The main roof is fractured to form rock blocks A, B and C, and the interaction between the rock blocks forms a hinge structure. Rock block A is still supported by the immediate roof, which is relatively stable. Rock block C is supported by the gangue on the side of the goaf, and its stability is poor. Both ends

of rock block B are supported by the immediate roof and gangue in the goaf, respectively, and rotate towards the goaf around the elastic-plastic boundary of solid coal. The following assumptions are made for the mechanical model of surrounding rock: (1) there is no interaction between rock block B and C and the gangue at the side of the goaf; (2) the shear force between rock strata such as immediate roof and main roof is ignored; (3) the supporting capacity of coal body in the lateral plastic zone of the retaining roadway is not considered; (4) the supporting force at the side of the roadway is neglected. The structural mechanical model established according to the assumed conditions is shown in Figure 2.

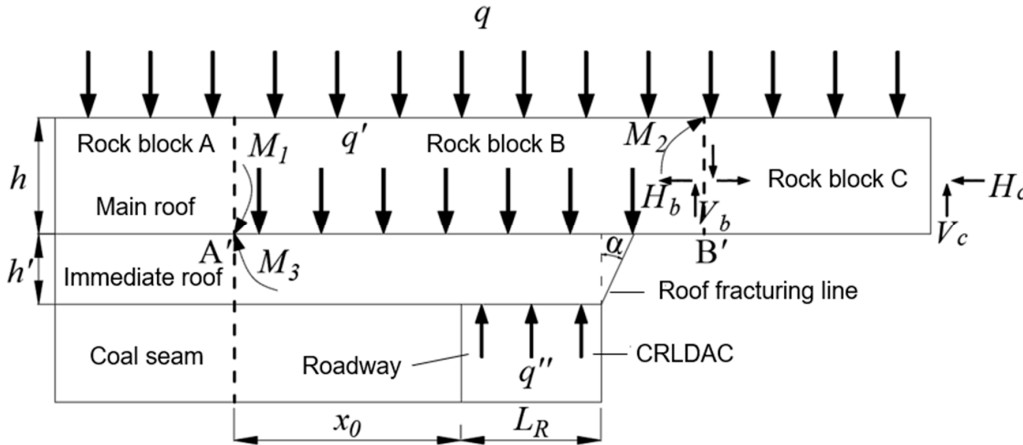

**Figure 2.** Mechanical model of roof structure of GERRCP.

The above picture is explained as follows. The key parameters of the system are as follows. After the rock strata fracture, the fracture length of key block B formed is [27]:

$$L = L_s \left( \sqrt{\frac{L_s^2}{L_f^2} + \frac{3}{2}} - \frac{L_s}{L_f} \right) \tag{1}$$

where, $L$: length of rock block; $L_s$: weighting interval of the immediate roof; $L_f$: working face length.

The horizontal force on rock block B is

$$H_b = \frac{qL}{2(h - S_b)} \tag{2}$$

where, $H_b$: horizontal force on rock block B; $q$: uniform load acting on the main roof; $h$: the thickness of basic roof; and $S_b$: the subsidence of rock block B.

The research on the mechanism of arch effect and its boundary conditions was studied, and the calculation of the plastic zone was referenced, and the width of the stress limit equilibrium zone in coal body was obtained [27,28].

$$x_0 = \frac{h_r A}{2tan\varphi_0} ln \left( \frac{k\gamma H + \frac{c_0}{tan\varphi_0}}{\frac{c_0}{tan\varphi_0} + \frac{p_x}{A}} \right) \tag{3}$$

where, $x_0$: width of stress limit equilibrium zone in coal body; $h_r$: roadway height; $A$: lateral pressure coefficient; $C_0$: cohesion of the interface between coal and rock; $\varphi_0$: internal friction angle; $K$: stress concentration factor; $\gamma$: average bulk density of overburden; $H$: roadway burial depth; $P_x$: support resistance of coal sides.

After mining, the main roof structure breaks under periodic pressure, forming a hinge structure, and the structure is in equilibrium. Through static analysis of hinge structure under this equilibrium condition, the static equilibrium equations of rock block C and B are established, such as Formulas 4 and 5. Among them, the support force provided by the support body in the roadway is simplified

as the load collection degree of support, and the support load in the roadway is solved based on the above analysis.

(1) Rock block C:

$$\Sigma X = 0, H_b - H_c = 0 \tag{4}$$

$$\Sigma Y = 0, qL + V_b - V_c = 0 \Sigma M_{B'} = 0, -M_2 + H_c(h/2 - S_c) + V_c L - H_b(h/2 - S_c) - qL^2/2 = 0$$

where, $H_c$: horizontal force on rock block C; $V_c$: the vertical force on rock block C; $V_b$: the vertical force on rock block B; $M_2$: moment of rock block C at section B; and $S_c$: the subsidence of rock block C.

(2) Rock block B:

$$\Sigma M_{A'} = 0, -M_1 - M_3 + q'' L_R(x_0 + L_R/2) + H_b(h/2 - S_b) - qL^2/2 + M_2 + V_b L + q'(x_0 + L_R + h' \tan\alpha)$$
$$\left[(L_R + x_0)^2 + (x_0 + L_R + h' \tan\alpha)^2 + (L_R + x_0)(x_0 + L_R + h' \tan\alpha)\right]/3(2x_0 + 2L_R + h' \tan\alpha) = 0 \tag{5}$$

where, $M_1$: moment of rock block B at section A; $M_3$: moment of immediate roof to basic roof; $q''$: load collection degree of support in the roadway; $L_R$: roadway width; $q'$: the uniform load acting on the immediate roof; $h'$: the thickness of immediate roof; and $\alpha$: pre-cracking roof cutting angle.

(3) The load collection degree of support in the roadway can be obtained simultaneously.

$$q'' = \left\{ \begin{array}{c} M_1 + M_3 - 2M_2 + qL^2 - qL(h - 2S_b)/4(h - S_b) - q'(x_0 + L_R + h' \tan\alpha) \\ \left[(2x_0 + 2L_R + h' \tan\alpha)^2 - (x_0 + L_R)(x_0 + L_R + h' \tan\alpha)\right]/3(2x_0 + 2L_R + h' \tan\alpha) \end{array} \right\} / L_R(x_0 + L_R/2) \tag{6}$$

Based on the new technology of GERRCP, the mechanical model of the roof structure is established. Through the mechanical analysis of the model, the key parameters such as fracture length of key blocks in upper strata and horizontal force acting on it are introduced, and the extension depth of the plastic zone in the solid coal side of the roadway under this technical condition is obtained, which provides certain theoretical support for the support design of the solid coal side of the mining roadway. In addition, through the static analysis of the balanced hinge structure under the condition of GERRCP, the corresponding static equilibrium equation is established, and the support load in roadway under this condition is obtained, which provides corresponding theoretical support for the support problem of mining roadway.

## 2.2. Technical Process of GERRCP

The mining mode layout of the traditional "121" construction method is shown in Figure 3a. When the mining system is used for coal mining, pillars are left, which belongs to the mining method of "one working face and two roadways". Different from the traditional mining method, the GERRCP is shown in Figure 3b, which is a typical single working face and single roadway without a pillar mining method. That is to say, the up and down drifts on the first face should be excavated firstly, and then at the same time in the working face of the mining, the retaining roadway, as the transport roadway on the next working face, is formed through the reinforcement of the advance anchor cable, pre-cracking roof cutting and gangue support at the side of the goaf. Therefore, the mining ratio is reduced and non-pillar mining is realized.

The technological process of GERRCP is shown in Figure 4. Its core can be summarized as four steps: strengthening, cutting, protecting and closing, that is: (1) adopt the CRLDAC to actively strengthen the supporting roadway roof according to the designed supporting parameters (Figure 4a); (2) the energy-gathered pre-cracking blasting hole shall be constructed at a certain distance in advance of the working face, and the bidirectional energy-gathered pre-cracking blasting shall be carried out according to the parameters determined by the blasting test to form a slit on the roof at the side of the goaf (Figure 4b); (3) after mining at the working face, the sliding and yielding gangue support structure and the constant pressure retractable support equipment are adopted behind the working face to strengthen the support in time. Under the action of the underground pressure, the roof at the side of the goaf collapses along the structural surface of the roof cutting slit to form the side of a new

roadway (Figure 4c); (4) the caving roof is gradually compacted as the working face advances, and the side of the roadway formed by caving is shotcreted to close the goaf. After the roadway is stabilized, the temporary support equipment is removed to realize the retaining new roadway (Figure 4d).

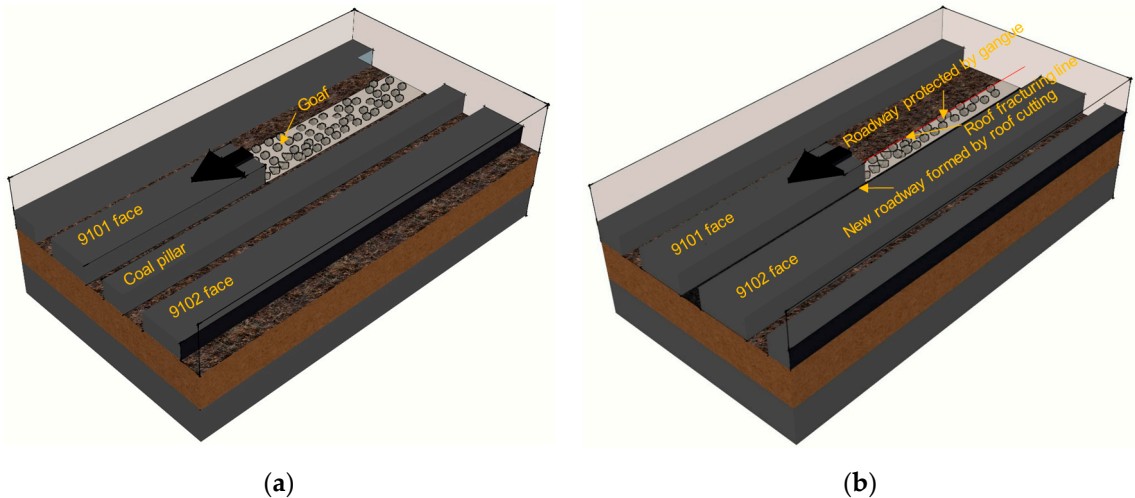

(**a**)                    (**b**)

**Figure 3.** Layout of mining mode. (**a**) Layout of traditional mining mode; (**b**) Layout of non-pillar mining mode.

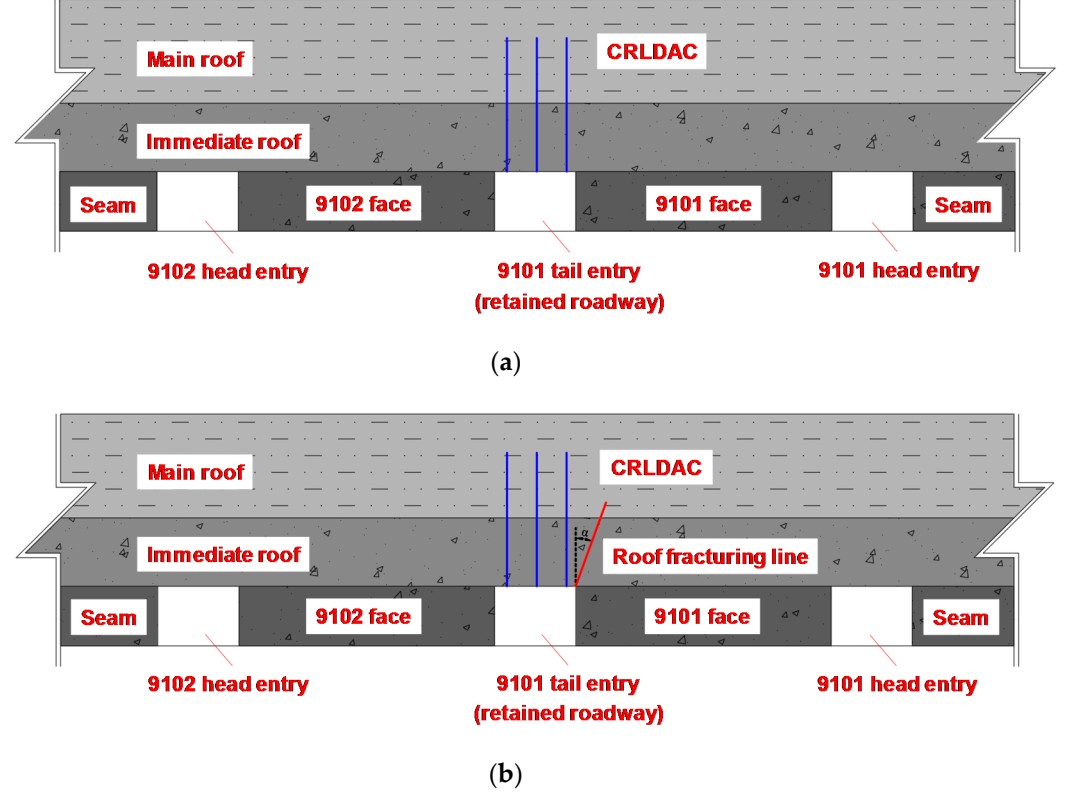

(**a**)

(**b**)

**Figure 4.** *Cont.*

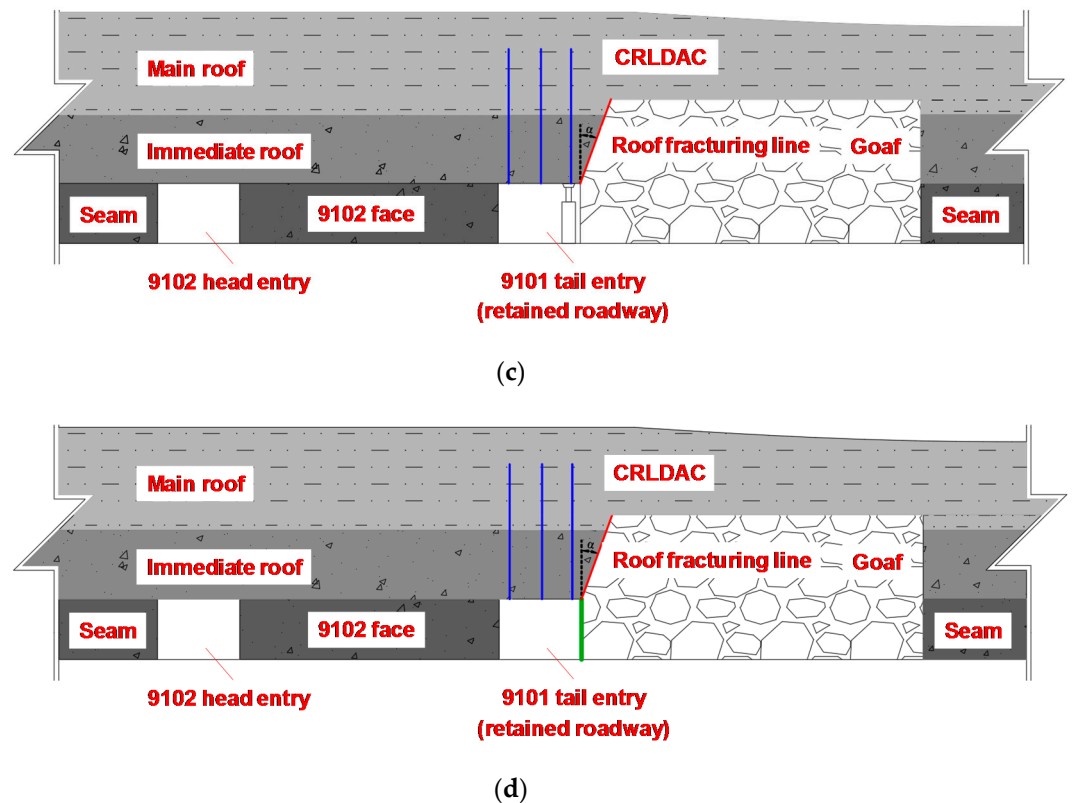

**Figure 4.** Technological process of GERRCP. (**a**) Strengthening roadway roof by constant-resistance large-deformation anchor cable (CRLDAC); (**b**) Pre-cracking roof cutting by energy-gathered blasting; (**c**) Gangue support; (**d**) Closing the goaf by shotcreting.

## 3. Engineering Background

The coal seam in the 9101 working face of Xiashanmao coal mine is located in the lower part of Taiyuan Formation. The thickness of the coal seam is 1.55–3.5 m, and the designed mining height is 3 m, which belongs to the medium-thick coal seam mining face. The dip angle of coal seam is 2–8° and the buried depth is 180–260 m. The distance between 8 # coal seam and 9 # coal seam is 11.20–19.60 m, with an average of 15.24 m. The immediate roof is mudstone, with an average thickness of 4.1 m; the basic roof is mainly sandy mudstone with an average thickness of 8.0 m, according to the drilling measurement on the roof and floor of the working face. The immediate and basic bottoms are mudstone and fine sandstone, respectively, as shown in Table 1.

**Table 1.** Lithological characteristics of roof and floor of coal seam.

| Name of Roof and Floor | Lithology | Thickness/m | Feature Description |
|---|---|---|---|
| Main roof | Sandy mudstone | 4.6–9.1 | Grey, block structure |
| Immediate roof | Mudstone | 3.6–4.2 | Black, block structure |
| 9 # coal seam | Coal seam | 2.8–3.1 | Black, vitreous luster, occurrence stability |
| Immediate bottom | Mudstone | 4.8–8.5 | Grey, block structure |
| Basic bottom | Fine sandstone | 13.2–26.6 | Grey, block structure, horizontal joint |

The test face is the first working face of the first mining area of 9 # coal seams, with a strike length of 480 m and an inclination of 150 m (as showed in Figure 5). The roof is managed by all caving method,

which adopts full-seam, comprehensive mechanized, and retreating mining methods. The average 12 m above the 9101 working face is the 8102 working face, which serves as the mining protective layer of 9101 working face. Among them, the 8102 working face adopts the "121" construction method of longwall mining, and the 9101 working face adopts the self-formed roadway without a pillar-mining system, and the mining direction is vertical intersection. The adjacent face is the 9102 working face, which is located at the south of the 9101 working face. The test roadway is the ventilation roadway of the 9101 working face. The roadway section is rectangular, with a width of 4000 mm and a height of 3200 mm.

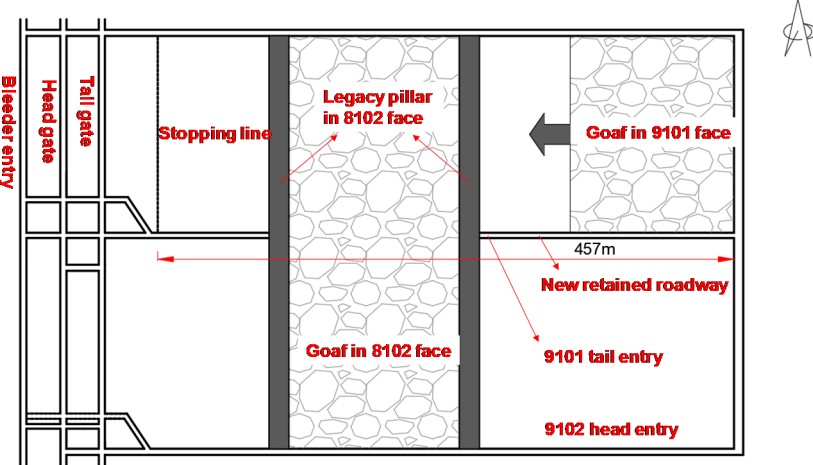

**Figure 5.** Layout of test working face of GERRCP in Xiashanmao coal mine.

## 4. Numerical Calculation and Analysis

### 4.1. Construction of Numerical Model

According to the specific engineering geological conditions of the 8102 and 9101 working faces of Xiashanmao coal mine, and combined with the existing underground pressure monitoring results, the finite difference software FLAC3D was used to establish a three-dimensional solid model. The distribution characteristics of mining stress in the mining process of the 9101 working face were studied. The numerical calculation model is shown in Figure 6. The calculation range was 330 m × 230 m × 100 m (length × width × height). The model simulated 12 layers of strata, including 8 # coal seam, 9 # coal seam and roof and floor strata, and truly reflected their occurrence conditions. Because the coal seam under actual working conditions could be regarded as a near-horizontal coal seam, the coal seam in the model was designed as a horizontal coal seam.

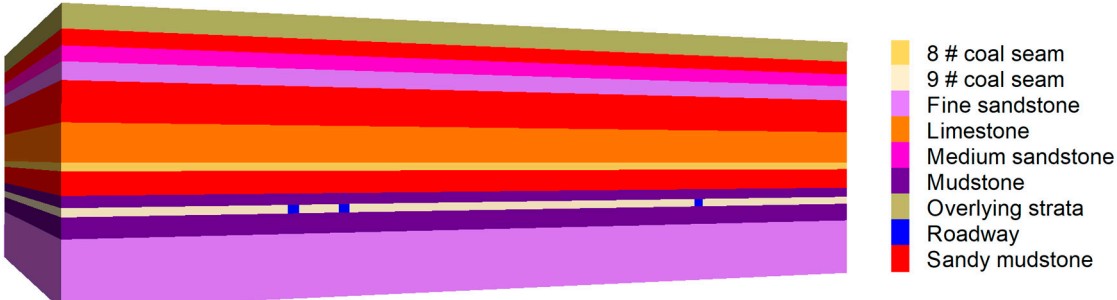

**Figure 6.** Three-dimensional numerical calculation model.

### 4.2. Determining Model Parameters

For rock mass materials, the elastic modulus of rock mass greatly influences the accuracy of simulation results. Therefore, in the process of numerical simulation, the rock elastic modulus should be corrected and verified repeatedly to reduce the error with the actual value to ensure the reliability of simulation results. The physical and mechanical parameters of strata were obtained according to the test of rock core samples from geological drilling in the testing field, and the elastic modulus of rock was taken to be 1/10 of the elastic modulus of rock block by comparing with the physical parameters of rock strata in the adjacent working face. In this study, the mole-coulomb model was selected as the constitutive model, and the effective physical and mechanical parameters of rock mass were finally determined based on the physical and mechanical parameters of the rock mass involved, as shown in Table 2.

**Table 2.** The physical and mechanical parameters of rock.

| Lithology | Density (kg·m$^{-3}$) | Bulk Modulus (GPa) | Shear Modulus (GPa) | Cohesion (MPa) | Internal Friction Angle (°) | Tensile Strength (MPa) |
|---|---|---|---|---|---|---|
| Upper strata | 2620 | 6.47 | 4.09 | 1.61 | 35 | 0.82 |
| Sandy mudstone | 2512 | 12.36 | 7.21 | 2.04 | 33 | 0.74 |
| Medium Sandstone | 2670 | 23.46 | 15.20 | 4.45 | 40 | 5.14 |
| Fine sandstone | 2870 | 21.04 | 13.52 | 3.20 | 42 | 1.29 |
| Sandy mudstone | 2503 | 10.63 | 5.59 | 2.04 | 33 | 0.74 |
| Limestone | 2910 | 29.26 | 18.27 | 5.14 | 42 | 7.31 |
| 8 # coal seam | 1380 | 4.91 | 2.01 | 1.25 | 32 | 0.15 |
| Sandy mudstone | 2531 | 14.13 | 9. 18 | 4.35 | 33 | 0.81 |
| Mudstone | 2488 | 9.97 | 7.35 | 1.20 | 32 | 0.58 |
| 9 # coal seam | 1450 | 4.91 | 2.01 | 1.25 | 32 | 0.15 |
| Mudstone | 2460 | 5.12 | 4.73 | 1.20 | 32 | 0.58 |
| Fine sandstone | 2870 | 21.04 | 13.52 | 3.75 | 38 | 1.84 |

### 4.3. Simulation Scheme

According to the actual working conditions, the corresponding simulation scheme was formulated. The geometry and boundary conditions of the model are shown in Figures 7 and 8 respectively. The model was fixed around to limit the horizontal movement, and fixed at the bottom to limit the vertical movement, and 5 MPa uniform load was applied at the top to simulate the self-weight boundary of the overlying strata. The mining scheme of coal seam was as follows. For the large pillar mining, (a) the 8102 working face was firstly mined step by step, the excavation footage of each step was set as 10 m. After each step was balanced, the next step of excavation was calculated to balance, and the calculation was carried out step by step. (b) Three mining roadways of 9101 and 9102 working faces were excavated at one time, and a pillar with a width of 15 m was set, and the calculation was made to the model balance. (c) The 9101 working face was excavated step by step, and the excavation footage was set as 10 m. The mining direction was perpendicular to that of the 8102 working face. For the small pillar mining, (a) ditto; (b) three mining roadways of the 9101 and 9102 working faces were excavated at one time, a pillar with a width of 5 m was set, and the calculation was made to the model balance; (c) ditto. For the non-pillar mining, (a) ditto; (b) two mining roadways of the 9101 working face were excavated at one time, and the calculation was made to the model balance; (c) pre-cracking roof cutting was conducted firstly. Then the 9101 working face was excavated step by step, and the excavation footage was set to 10 m. The mining direction was perpendicular to that of the 8102 working face.

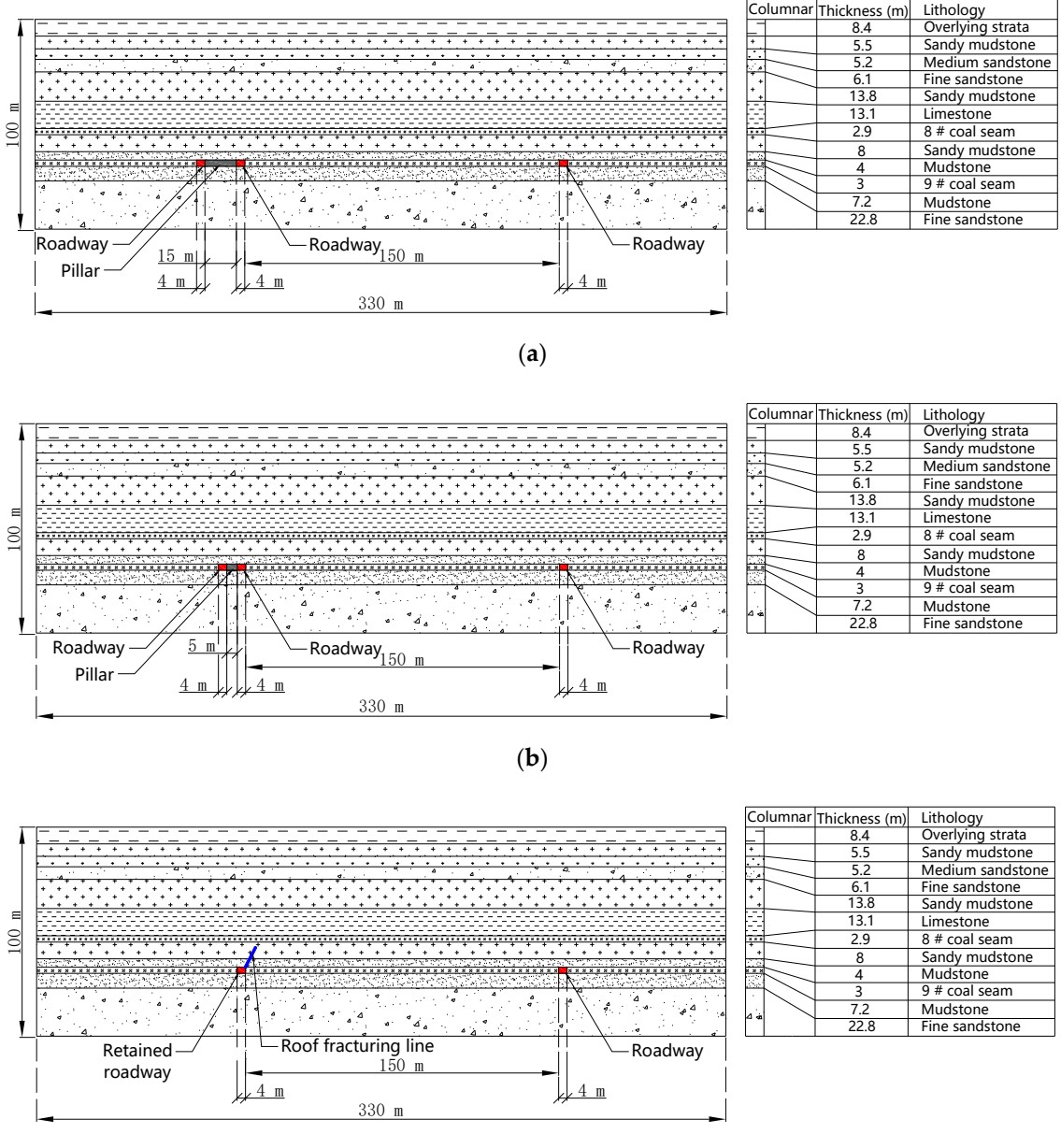

**Figure 7.** Geometry of the model. (**a**) Geometry of model of large pillar mining; (**b**) geometry of model of small pillar mining; (**c**) geometry of model of non-pillar mining.

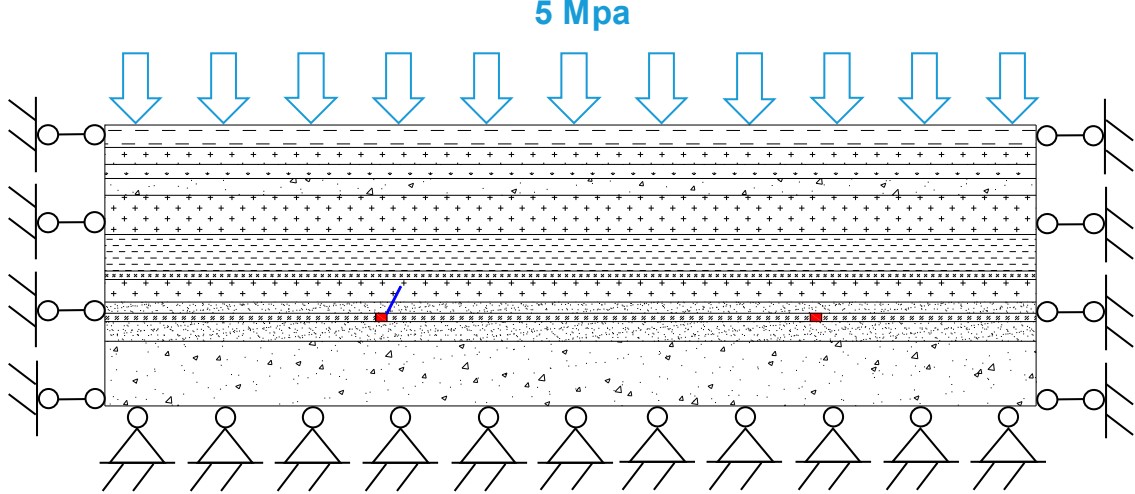

**Figure 8.** Boundary conditions of the model.

## 5. Distribution Law of Three-Dimensional Stress Field

### 5.1. Distribution Law of Vertical Stress

In order to analyze the evolution law of stress distribution in the surrounding rock of stope, when the working face advances to 120 m, monitoring lines are arranged at different positions of the model, and vertical stress is monitored by monitoring lines. The location of monitoring lines is shown in Figure 9. Among them, one, two and three monitoring lines are located in the inner 10 m of the 9101 ventilation roadway, the middle of the 9101 working face, and the inner 10 m of the 9101 haulage roadway; four, five and six monitoring lines are located 5 m, 10 m and 20 m in front of the working face, respectively; seven, eight and nine monitoring lines are located on the left side of the working face, 5 m, 10 m and 15 m in front of the working face respectively; 10, 11, 12 and 13 monitoring lines are located on the left side of working the face, 5 m, 10 m, 30 m, and 80 m behind the working face, respectively.

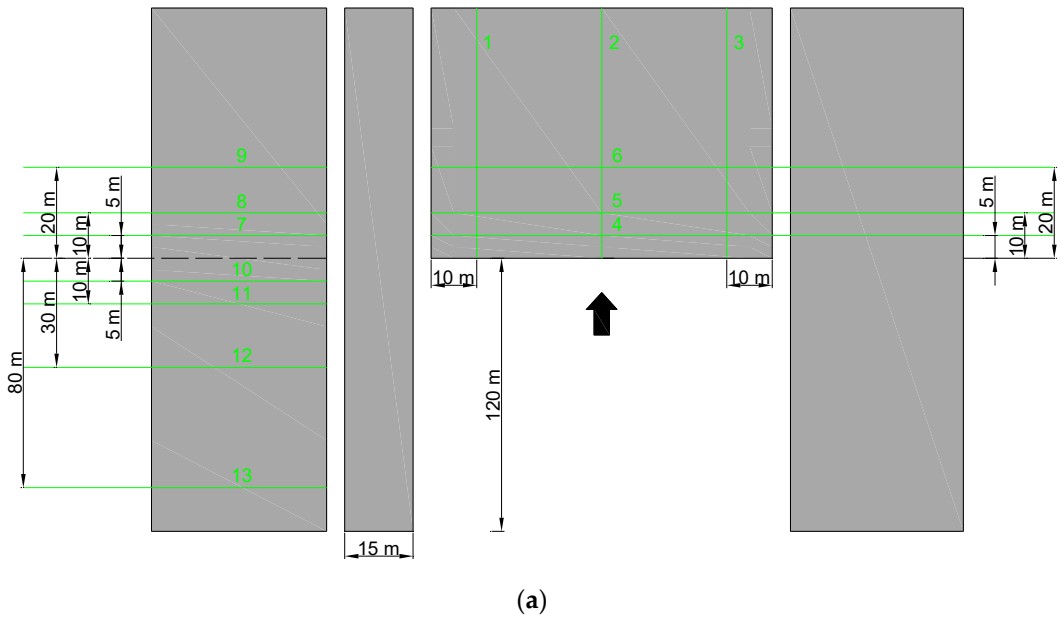

(**a**)

**Figure 9.** *Cont.*

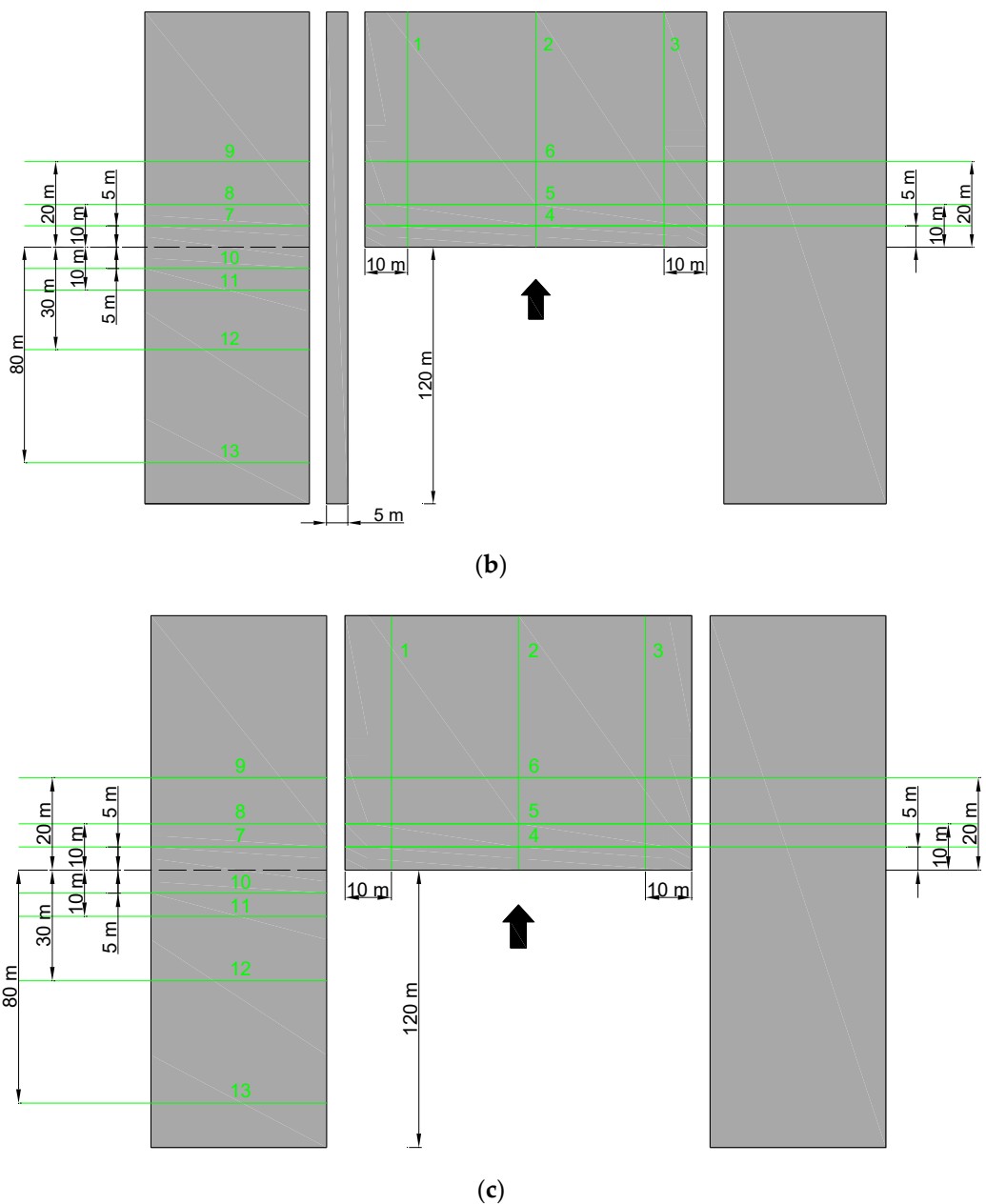

**Figure 9.** Location of monitoring lines. (**a**) Location of monitoring lines of large pillar mining; (**b**) Location of monitoring lines of small pillar mining; (**c**) Location of monitoring lines of non-pillar mining.

5.1.1. Stress Distribution in Front of Working Face

(1)  Stress distribution along working face strike

The vertical stress distribution nephograms and stress distribution curves at monitoring line 1 are shown in Figure 10. Figure 10 shows that the vertical stress distribution along the working face strike was similar under the three mining methods, and the advanced mining stress concentration area of the remaining pillars mining was larger than that of non-pillar mining, and the stress value was slightly higher. The peak of advanced mining stress was located 10 m in front of the working face, which was about 3.3 times that of the mining height of the working face. Among them, the peak stress of non-pillar mining was 14% lower than that of large pillar mining and 10% lower than that of small pillar mining. Within 3 m from the face, was the pressure-released zone of the stope; the vertical stress value was lower than the original rock stress, and the coal body mainly underwent

plastic deformation. Under the coupling effect of the overlying goaf and mining abutment pressure of this coal seam, a pressure boost belt was formed within the range of 3–33 m from the working face. The elastic deformation of coal body in this area caused the accumulation of elastic deformation energy, and the bearing capacity was higher. Within the range of 33–90 m from the working face, under the influence of pressure relief in the goaf of 8 # coal seam, the stress value was low. From 90–110 m away from working face, the residual pillar of 8 # coal seam formed a stress concentration zone here, and the stress value rose again. The stress distribution law at monitoring lines 2 and 3 was similar to that at monitoring line 1. The peak of advance mining stress was 10 m ahead of the working face. The peak stress of non-pillar mining was 23.88 MPa and 19.92 MPa respectively, which were 8%, 7% and −1%, 1% lower than that of traditional mining respectively (as shown in Table 3).

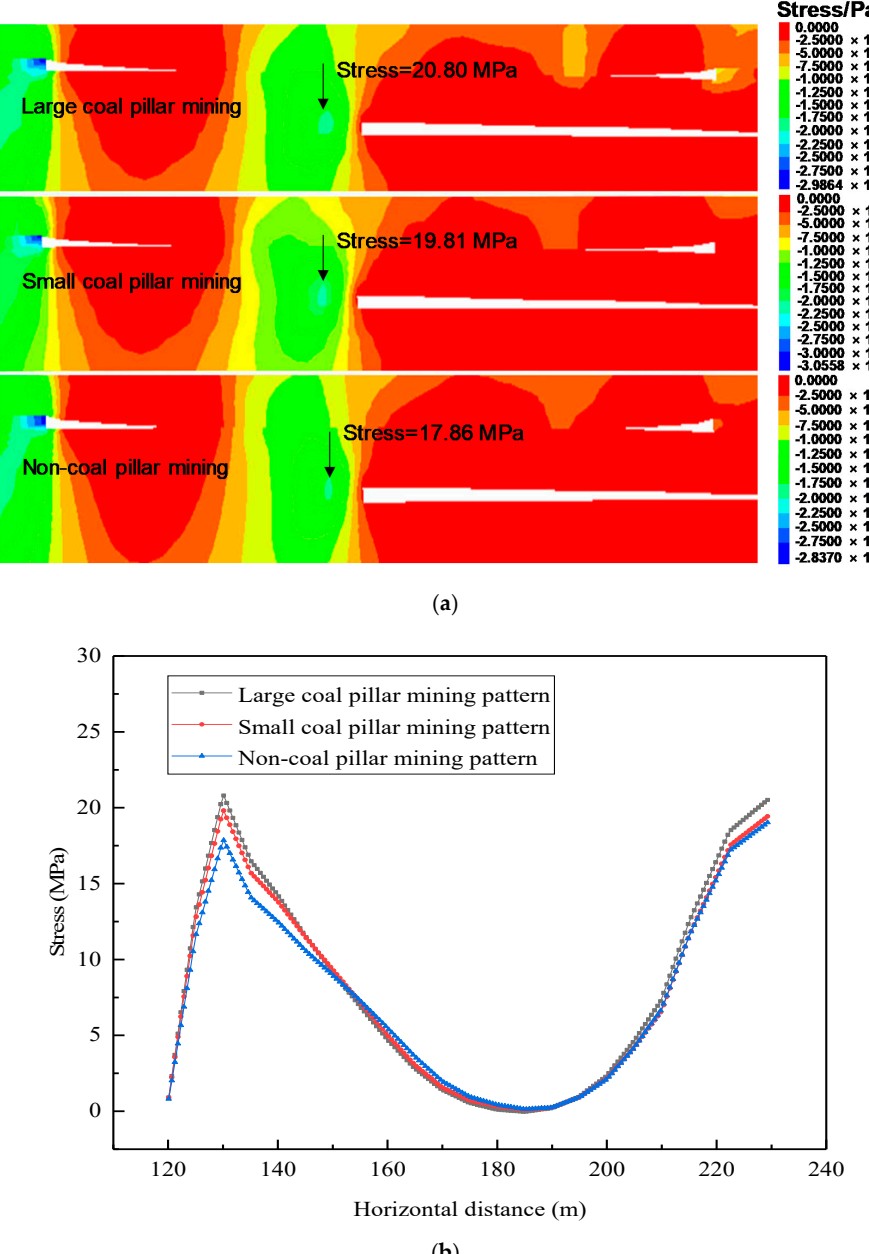

**Figure 10.** Vertical stress nephogram and stress distribution curve at the monitoring line 1 along the working face strike. (**a**) Vertical stress nephogram; (**b**) Stress distribution curve.

**Table 3.** Key parameters of the stress distribution characteristics along the working face strike.

| Monitoring Line | Mining Mode | Stress Trend | Peak Position (m) | Peak Size (MPa) | Peak Reduction of (%) |
|---|---|---|---|---|---|
| 1 | Large | Similar | 10 | 20.80 | 14 |
| | Small | | 10 | 19.81 | 10 |
| | GERRCP | | 10 | 17.86 | Reference quantity |
| 2 | Large | Similar | 10 | 26.05 | 8 |
| | Small | | 10 | 25.63 | 7 |
| | GERRCP | | 10 | 23.88 | Reference quantity |
| 3 | Large | Similar | 10 | 19.64 | −1 |
| | Small | | 10 | 20.05 | 1 |
| | GERRCP | | 10 | 19.92 | Reference quantity |

Based on the above analysis, the key parameters of the stress distribution characteristics along the working face strike in the simulation results were summarized, and a matrix-type chart with resulting values and illustrations was made, so they can be visualized and compared in a single view (as shown in Table 3).

(2)   Stress distribution along the inclination of working face

The vertical stress distribution nephograms and stress distribution curves at monitoring line 4 are shown in Figure 11. Figure 11 showed that, the distribution law of vertical stress along the inclination of the working face was similar under the three mining modes, that was, the stress data showed that the vertical stress increased first and then decreased. The stress curves of large and small pillars basically coincided, and the stress value was slightly higher than that of the non-pillar mining method. At the edge of the ventilation roadway, the three stresses were 10.8 MPa, 10.2 MPa and 7.6 MPa, respectively. The stress of non-pillar mining at the edge of the ventilation roadway was 30% and 25% lower than that of the large pillar and small pillar, respectively. At the distance of 10 m from the ventilation roadway, the three stresses reached 12.8 MPa, 12.3 MPa and 11.1 MPa. The stress of the non-coal pillar mining was 13% and 10% lower than that of the large pillar and small pillar mining, respectively. It could be seen that within the distance of 10 m from the ventilation roadway, the vertical stress increased greatly, while the non-coal pillar mining showed the characteristics of low stress compared with the traditional mining method. The main reason was that the roof rock beam was cut off by pre-cracking the roof cutting, which transformed it from a long-wall beam to short-wall beam and cut off the stress transfer between roofs, which improved the stress condition of surrounding rock obviously. The stress distribution law at monitoring lines 5 and 6 was similar to that at monitoring line 4. The stress increased rapidly within 10 m from the ventilation roadway and then slowed down. The stress distribution of traditional mining was axisymmetrical about the middle of the working face, while the stress of the side of the roof cutting of non-pillar mining was significantly lower than that of the non-roof cutting. Among them, the peak stress of non-pillar mining was 23.98 MPa and 19.41 MPa respectively, which were 7%, 7% and 10%, 8% lower than that of traditional mining respectively (as shown in Table 4).

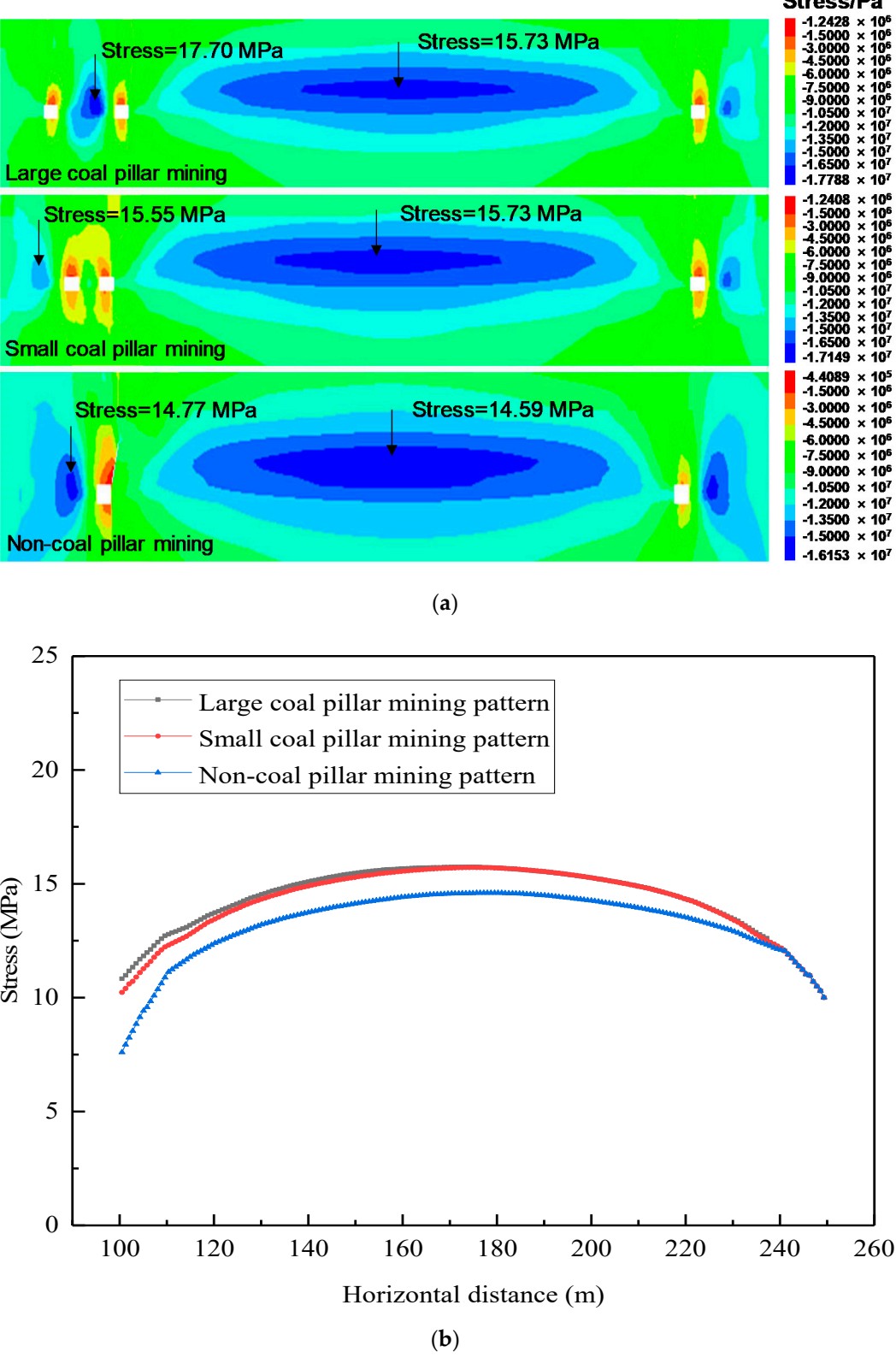

**Figure 11.** Vertical stress nephogram and stress distribution curve at the monitoring line 4 along the inclination of the working face. (**a**) Vertical stress nephogram; (**b**) Vertical stress distribution curve.

**Table 4.** Key parameters of the stress distribution characteristics along the inclination of the working face.

| Monitoring Line | Mining Mode | Stress Concentration around Roadway (MPa) | Peak Reduction (%) | Stress Concentration of Working Face (MPa) | Peak Reduction (%) |
|---|---|---|---|---|---|
| 4 | Large | 17.70 | 17 | 15.73 | 7 |
| | Small | 15.55 | 5 | 15.73 | 7 |
| | GERRCP | 14.77 | Reference quantity | 14.59 | Reference quantity |
| 5 | Large | 15.34 | 12 | 25.87 | 7 |
| | Small | 14.60 | 8 | 25.79 | 7 |
| | GERRCP | 13.50 | Reference quantity | 23.98 | Reference quantity |
| 6 | Large | 14.45 | 13 | 21.60 | 10 |
| | Small | 13.40 | 7 | 21.18 | 8 |
| | GERRCP | 12.50 | Reference quantity | 19.41 | Reference quantity |

Through the above analysis, the key parameters of the stress distribution characteristics along the inclination of the working face were summarized, and a matrix-type chart with resulting values and illustrations was made (as shown in Table 4).

### 5.1.2. Stress Distribution in Lateral Direction of Working Face

(1)　Stress distribution in left front of working face

The vertical stress distribution curves at monitoring line 7 are shown in Figure 12. Figure 12 showed that there were two peaks along the working face inclination under the condition of large and small pillars mining. The locations of peak stress were 6 m, 23 m and 2 m, 15 m away from the edge of ventilation roadway respectively, with sizes of 17.7 MPa, 12.3 MPa and 10.8 MPa, 15.6 MPa. The stress concentration factors were 2.6, 1.8 and 1.6, 2.3. There was a wave peak along the inclination of the working face under non-pillar mining, which was located 5 m outside ventilation roadway, with a size of 14.8 MPa, and the stress concentration factor was 2.2. Compared with the peak stress, it could be seen that the non-pillar mining decreased by 16% and 5% for large and small pillar mining respectively, indicating that the non-pillar mining had less influence on the advanced mining stress of the roadway surrounding rock, that was, the technology of roof cutting and pressure relief cut off the stress transfer between the working face and roadway roof, and the stress control effect was better. By analyzing the location of peak stress, it could be seen that the peak stress of the retaining pillar mining mode was 6 m outside the roadway, while that of the non-pillar mining mode was 5 m outside the roadway, which indicated that the plastic zone of surrounding rock in the advanced position of a roadway under non-pillar mining mode was smaller. In addition, the advantages of this mining method on the stress control of surrounding rock were further reflected. The stress distribution law at monitoring lines 8 and 9 was similar to that at monitoring line 7. The advance abutment pressure in lateral direction of the working face was lower than that of monitoring line 7. The peak stress of traditional mining was 6 m outside the ventilation roadway, while the peak stress of non-pillar mining was 5 m from the ventilation roadway. The peak stresses of non-pillar mining were 13.5 MPa and 12.5 MPa respectively, which were 12%, 8% and 13%, 7% lower than that of traditional mining respectively (as shown in Table 5).

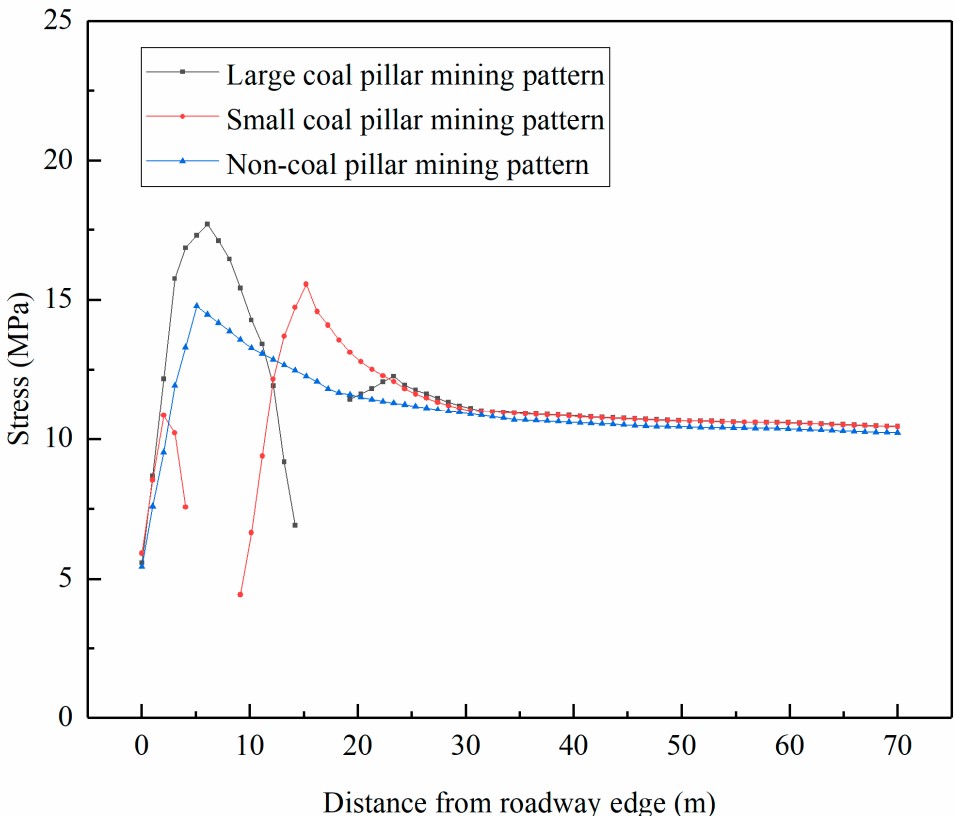

**Figure 12.** Vertical stress distribution curve at monitoring line 7 in lateral direction of the working face.

**Table 5.** Key parameters of the stress distribution characteristics in the left front of the working face.

| Monitoring Line | Mining Mode | Number of Peaks | Peak Position (m) | Peak Size (MPa) | Peak Reduction (%) |
|---|---|---|---|---|---|
| 7 | Large | 2 | 6 | 17.7 | 16 |
| | Small | 2 | 6 | 15.6 | 5 |
| | GERRCP | 1 | 5 | 14.8 | Reference quantity |
| 8 | Large | 2 | 6 | 15.3 | 12 |
| | Small | 2 | 6 | 14.6 | 8 |
| | GERRCP | 1 | 5 | 13.5 | Reference quantity |
| 9 | Large | 2 | 6 | 14.4 | 13 |
| | Small | 2 | 6 | 13.4 | 7 |
| | GERRCP | 1 | 5 | 12.5 | Reference quantity |

Combined with the analysis results, the key parameters of the stress distribution characteristics in the left front of the working face in the simulation results were summarized, and a matrix-type chart with resulting values and illustrations was made (as shown in Table 5).

(2) Stress distribution in left rear of working face

The vertical stress distribution curves at monitoring line 10 are shown in Figure 13. Figure 13 showed that, there were obvious differences in the stress distribution characteristics along the inclination of the working face under the three mining methods, which were mainly reflected in

the stress concentration area and the degree of stress concentration. Figure 13b intuitively showed that, in the lateral direction of the working face, the stress of the large pillar mining method was the largest, followed by the small pillar mining and the minimum of the non-pillar mining method. Under the traditional mining method, the vertical stress showed that the distribution characteristics were saddle-shaped, in which the large pillar mining first formed the stress concentration in the large pillar, that was, the first wave peak appeared. Then a small degree of concentration appeared at the roadway edge of the 9102 working face, forming a second wave peak, indicating that the stope underground pressure was mainly borne by the large pillar behind the working face. The small pillar mining was obviously different from large pillar mining. The larger stress concentration was located at the 9102 working face, and a smaller concentration occurred in the small pillar, indicating that the small pillar had a low bearing capacity behind the working face due to the limitation of coal pillar width, and the stress bearing area transferred to the deep part of the working face. Because of the elimination of pillars for the non-pillar mining method, the weight of overlying strata was borne by the solid coal on the working face, and a stress bearing area was formed in the 9102 working face. Therefore, the stress concentration position was transferred from the traditional pillar area to the deep part of the working face, and the peak of non-pillar mining was the smallest, which was 21% and 10% lower than the traditional mining method of large pillar and small pillar respectively. It could be seen that the technique of gob-side roof cutting effectively reduced the stress concentration in stope and optimized the stress distribution in the lateral direction of the working face. The stress distribution law at monitoring lines 11, 12, and 13 was similar to that at monitoring line 10. Traditional mining methods still showed the characteristics of double-stress wave peak distribution, in which the high stress concentration in large pillar, small pillar and no pillar mining methods occurred in the pillar, 9102 working face and deep part of the solid coal, respectively. The peak stresses were the smallest of non-pillar mining, which were 18%, 10%; 16%, 6% and 13%, 7% lower than that of traditional mining respectively (as shown in Table 6).

**Table 6.** Key parameters of the stress distribution characteristics in the left rear of the working face.

| Monitoring Line | Mining Mode | Number of Peaks | Peak Position (m) | Peak Size (MPa) | Peak Reduction (%) |
|---|---|---|---|---|---|
| 10 | Large | 2 | 6 | 21.52 | 21 |
| | Small | 2 | 6 | 18.80 | 10 |
| | GERRCP | 1 | 5 | 16.99 | Reference quantity |
| 11 | Large | 2 | 6 | 21.68 | 18 |
| | Small | 2 | 6 | 19.84 | 10 |
| | GERRCP | 1 | 5 | 17.83 | Reference quantity |
| 12 | Large | 2 | 6 | 23.70 | 16 |
| | Small | 2 | 6 | 21.17 | 6 |
| | GERRCP | 1 | 5 | 20.00 | Reference quantity |
| 13 | Large | 2 | 6 | 27.17 | 21 |
| | Small | 2 | 6 | 22.16 | 3 |
| | GERRCP | 1 | 5 | 21.58 | Reference quantity |

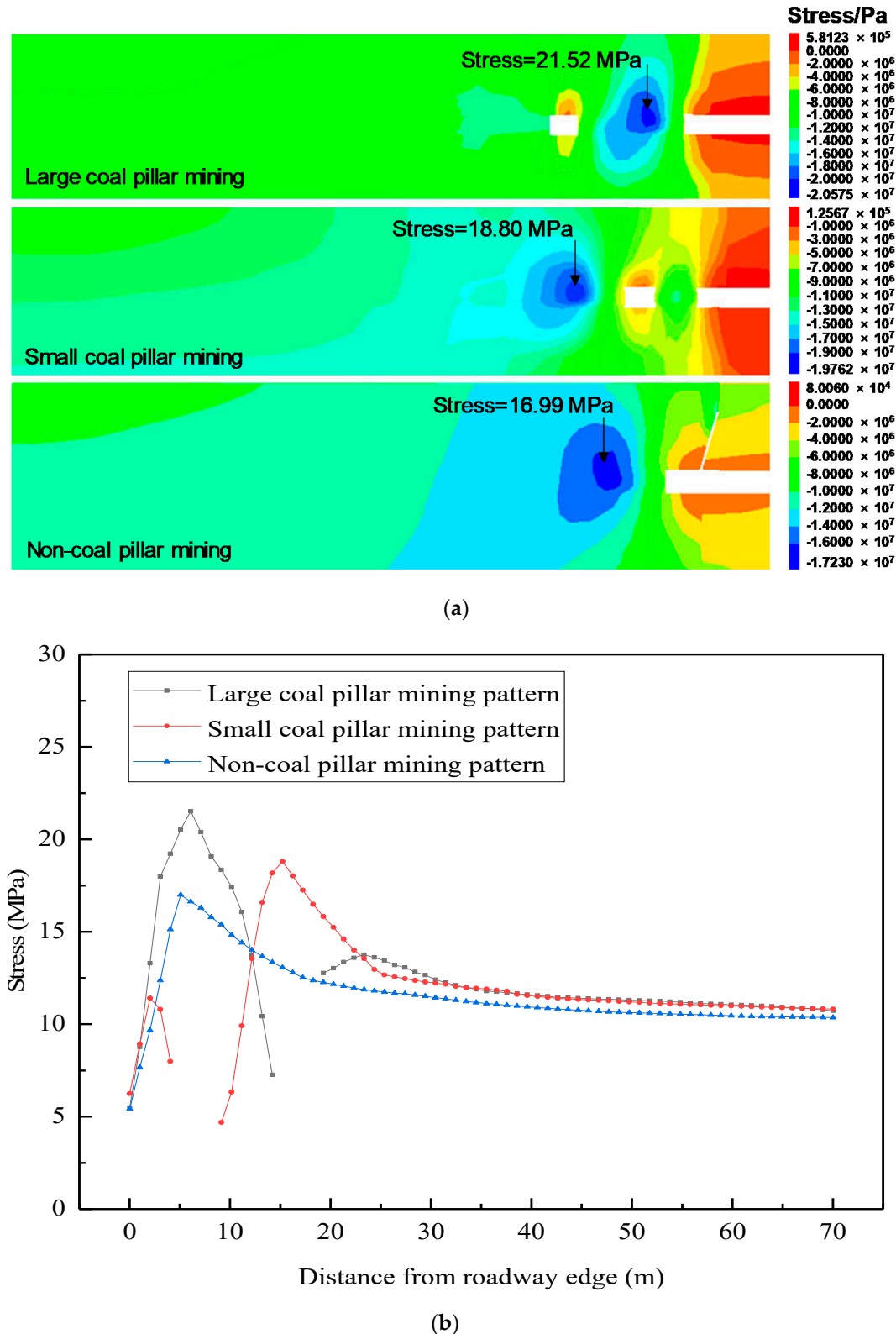

**Figure 13.** Vertical stress nephogram and stress distribution curve at monitoring line 10 in lateral direction of working face. (**a**) Vertical stress nephogram; (**b**) Vertical stress distribution curve.

With the help of the above analysis, the key parameters of the stress distribution characteristics in the left rear of the working face were summarized, and a matrix-type chart with resulting values and illustrations was made (as shown in Table 6).

## 5.2. Distribution Law of Horizontal Stress

In order to analyze the distribution of horizontal stress, the model was sliced horizontally. As this study focused on the analysis of the distribution characteristics of the mining stress field in 9 # coal seam, therefore, in the middle of the height range of the face, horizontal slices were made to analyze the evolution law of horizontal stress in the front and side of the working face.

### 5.2.1. Stress Distribution in Front of Working Face

When the working face advanced to 120 m, the horizontal stress distribution in the front and side of the working face under the three mining modes is shown in Figure 14. In front of the working face, the horizontal stress increased first and then decreased. Under the large pillar mining mode, the stress rising area was large with a span of about 31 m, and the peak stress was 20.5 MPa. The peak stress was located 10 m in front of the working face. Under the small pillar mining mode, the stress rising area was slightly smaller, the span was about 23 m, the peak stress was 16.8 MPa, and the stress wave peak was located 10 m ahead of the working face. The area of stress rise was the smallest under non-pillar mining, with a span of about 9 m, a peak stress of 15.2 MPa and the stress wave peak was located 10 m ahead of the working face.

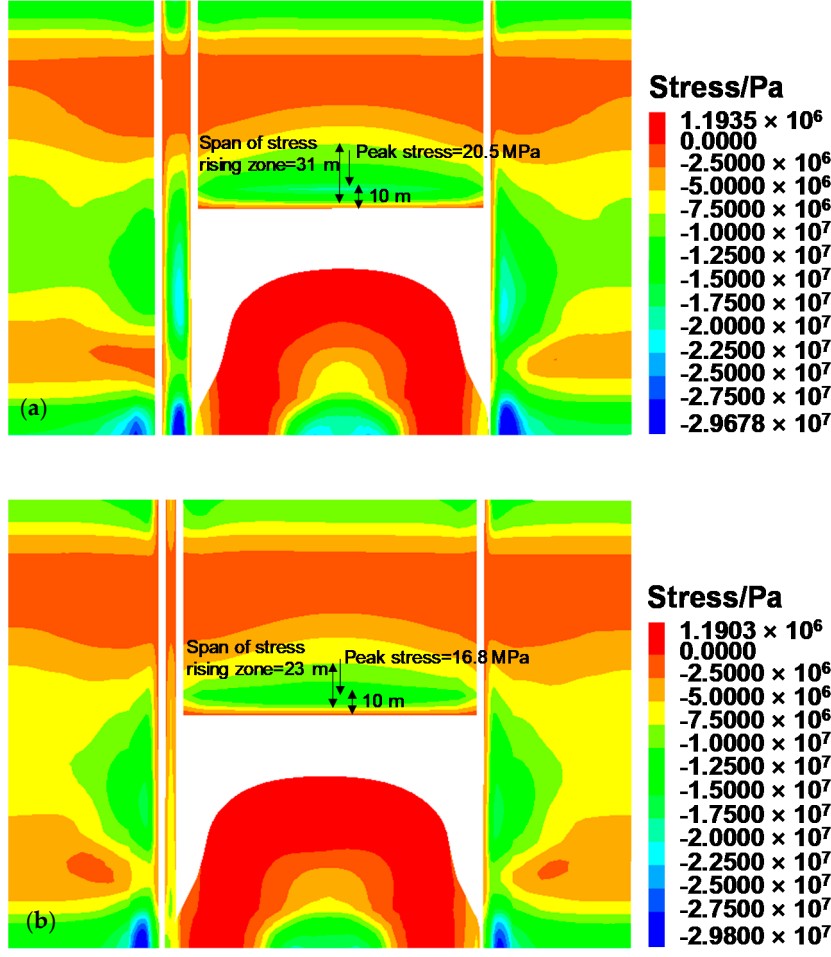

**Figure 14.** *Cont.*

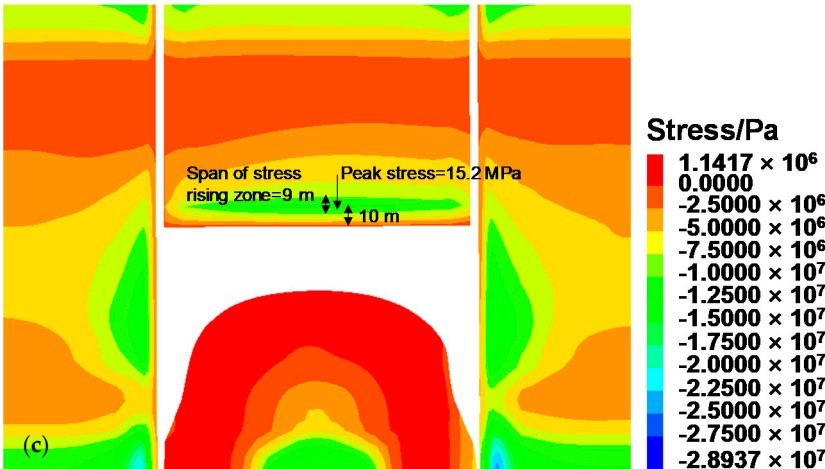

**Figure 14.** Horizontal stress distribution in front and side of the working face under the three mining modes. (**a**) Large pillar mining; (**b**) Small pillar mining; (**c**) Non-pillar mining.

The stress data showed that the non-pillar mining had the characteristics of a small stress rising area and small stress value. Therefore, it could be concluded that non-pillar mining technology could indeed improve the distribution of the surrounding rock stress field in front of the working face, which was of great significance to reduce the surrounding rock stress in the stope.

### 5.2.2. Stress Distribution in Lateral Direction of Working Face

Figure 14 showed that, in the lateral direction of the working face, high stress concentration was formed in the pillar under large pillar mining, and the stress concentration zone was also formed in the coal body of the 9102 working face; the stress concentration in the pillar under the small pillar mining was not obvious, but the phenomenon of a large stress concentration zone was formed in the 9102 working face; the phenomenon of stress rise occurred in the 9102 working face under non-pillar mining, but the area of the concentrated area was lower than that of the remaining pillars mining mode. This was because the retaining roadway by roof-cutting cut off the stress transfer between the working face and its lateral direction, so the value of the stress in the lateral direction of the working face was reduced and had the characteristics of optimizing the stress distribution in the lateral direction of the working face.

## 6. Mine Pressure Monitoring

### 6.1. Stress Monitoring

In former sections, the distributions of advance stress and lateral abutment pressure in 9101 working face were studied by numerical simulation. On-site measurements of the strike and lateral abutment pressure of 9101 working face were carried out in this section. With the help of monitoring data, the distribution law of mine pressure was analyzed to verify the numerical simulation results.

### 6.1.1. Monitoring System

At present, the KJ550 on-line stress monitoring system was used in Xiashanmao coal mine. The system consists of three main components (as shown in Figure 15): (1) The monitoring host and data processing and analysis system on the ground, adopting a high-performance integrated server workstation and high-performance computer, which can realize the functions of data storage and analysis. (2) The underground monitoring substation (including a power supply), whose main components are industrial high-performance electronic accessories. (3) The pressure sensing system arranged along the entry on both sides of the working face is composed of the borehole stress meter with hydraulic oil as the pressure-sensing medium and high-precision pressure sensors.

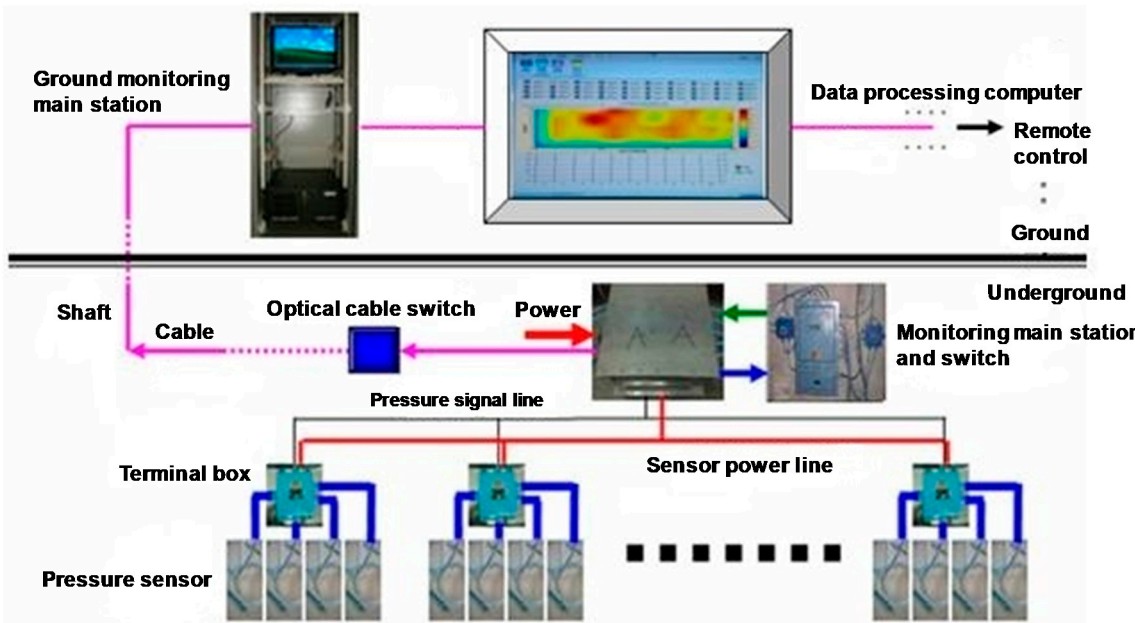

**Figure 15.** Schematic diagram of remote monitoring system.

The structure and working principle of the system are shown in Figure 15. Under the influence of mining, the pressure of coal and rock mass around the pressure sensor installed in the survey area changes. The sensor receives the pressure fluctuation signal and transmits it to the terminal box, which is transmitted through the pressure signal line to the monitoring main station and switch. The monitoring main station and switch convert the electrical signal into an optical signal, then transmit the optical signal to the ground-monitoring main station through the optical cable switch, and then transmit it to the data-processing computer for stress data processing, so as to realize remote control.

Real-time monitoring technology aims to detect and identify various potential abnormalities and faults, realizing real-time monitoring and early warning of dangerous situations, so as to take necessary measures for minimizing performance degradation and economic costs and avoid catastrophic situations [29–31]. Similarly, the KJ550 monitoring system can monitor the stress of coal body and rock mass in front of the working face and around the roadway in real time, and monitor and display the dynamic stress nephogram in front of the working face in real time, so as to realize the real-time monitoring and early warning of the hazardous area of rock burst. At the same time, it has the functions of remote control, data analysis and remote maintenance. Through the remote data processing and early warning center, the monitoring data can be analyzed and processed in real time.

### 6.1.2. Monitoring Programme

Figure 16 shows the layout of the 9101 working face and real-time monitoring system in Xiashanmao coal mine. It can be seen from the figure that two stations are arranged in the 9101 tail entry, and the measuring points of stations 1 and 2 are arranged in the solid coal side of the roadway to monitor the change of lateral abutment pressure of the coal side. There are four measuring points in stations 1 and 2, with buried depth of 3, 6, 9 and 12 m respectively; the distance between measuring points is 0.5–1 m, and the distance between measuring station 2 and station 1 is 100 m; Stations 3–22 are arranged in the 9101 working face to monitor the change of strike abutment pressure in working face, and each station is equipped with two measuring points with buried depths of 5 and 10 m. The distance between stations is 2 m and distance between measuring points is 0.5 m.

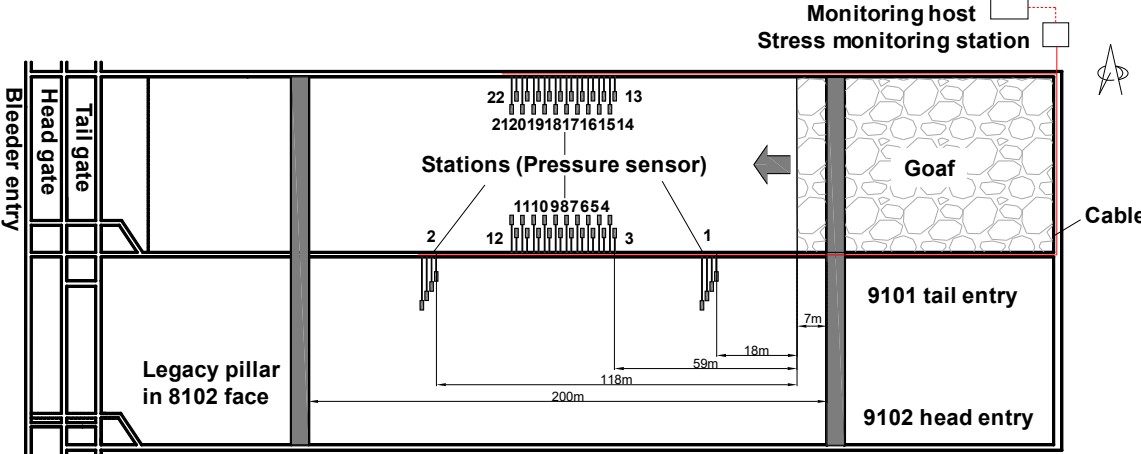

**Figure 16.** Layout of the 9101 working face and real-time monitoring system.

*6.2. Analysis of Main Monitoring Results*

6.2.1. Distribution Characteristics of Strike Abutment Pressure of Working Face

There are many stations for coal body layout in the inner side of drift in the 9101 working face. The monitoring results of two stations are selected for analysis below. Figure 17 is the relative vertical stress variation curve of each measuring point at station 3 of the lower drift and station 20 of the upper drift of the 9101 working face respectively.

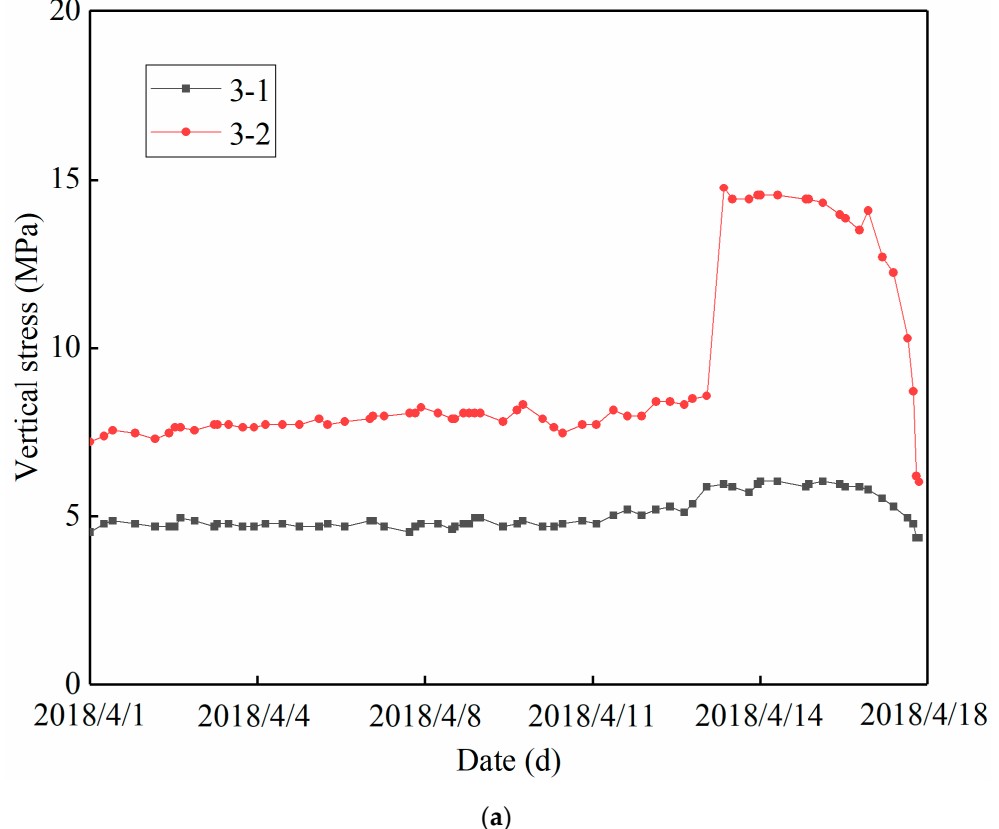

(**a**)

**Figure 17.** *Cont.*

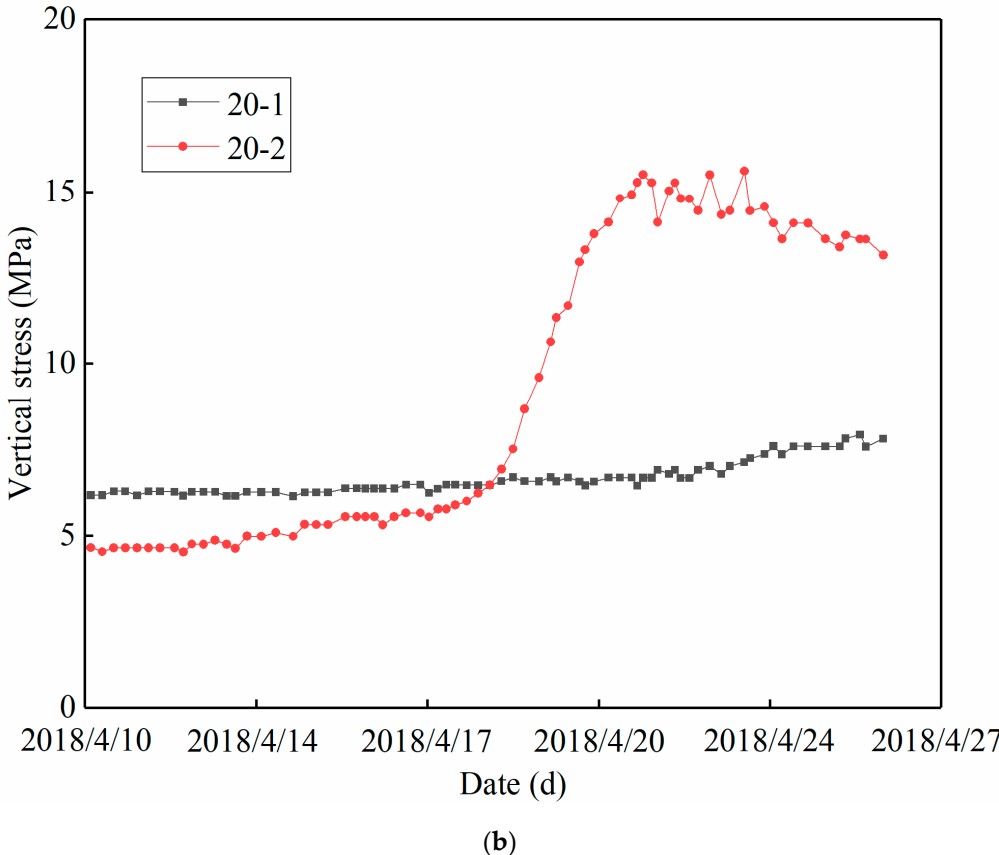

(**b**)

**Figure 17.** Relative vertical stress variation curve. (**a**) Relative vertical stress variation curve of station 3; (**b**) Relative vertical stress variation curve of station 20.

Figure 17a showed that, the vertical stress of the measuring point of the station increased significantly on 4/13, indicating that the measuring point began to enter the influence range of strike abutment pressure on the working face. At this time, the measuring point was 42.6 m away from the working face, that is, the influence range of strike abutment pressure on working face was 42.6 m. The vertical stress of the measuring point began to decrease on 4/17, indicating that the measuring point began to enter the plastic zone, which was 9 m away from the working face. It could be seen that the peak position of strike abutment pressure in the working face was 9 m away from the coal wall, and the continuous influence distance of abutment pressure was 33.6 m.

Figure 17b showed that the vertical stress of the 20-1 measuring point began to reach the peak abutment pressure on 4/20, when the measuring point was 54.2 m away from the working face, that is, the influence range of advance abutment pressure of the working face was 54.2 m. The vertical stress of the measuring point began to decrease significantly on 4/24, indicating that the measuring point has entered the plastic zone, which was 12 m away from working face, i.e., the peak position of the strike abutment pressure of the working face was 12 m away from the coal wall, and the continuous influence distance of the abutment pressure was 42.2 m.

6.2.2. Distribution Characteristics of Lateral Abutment Pressure of Working Face

Station 1 and Station 2 were continuously monitored for 45 days (Signal cable interruption in goaf at later stage). During this period, the 9101 working face pushed forward from 18 m in front of station 1 to 35 m after station 2, with a total of 153 m, and the distance of the signal cable entering the goaf was 60 m. Figure 18 showed the relative vertical stress variation curves of stations 1 and 2 respectively.

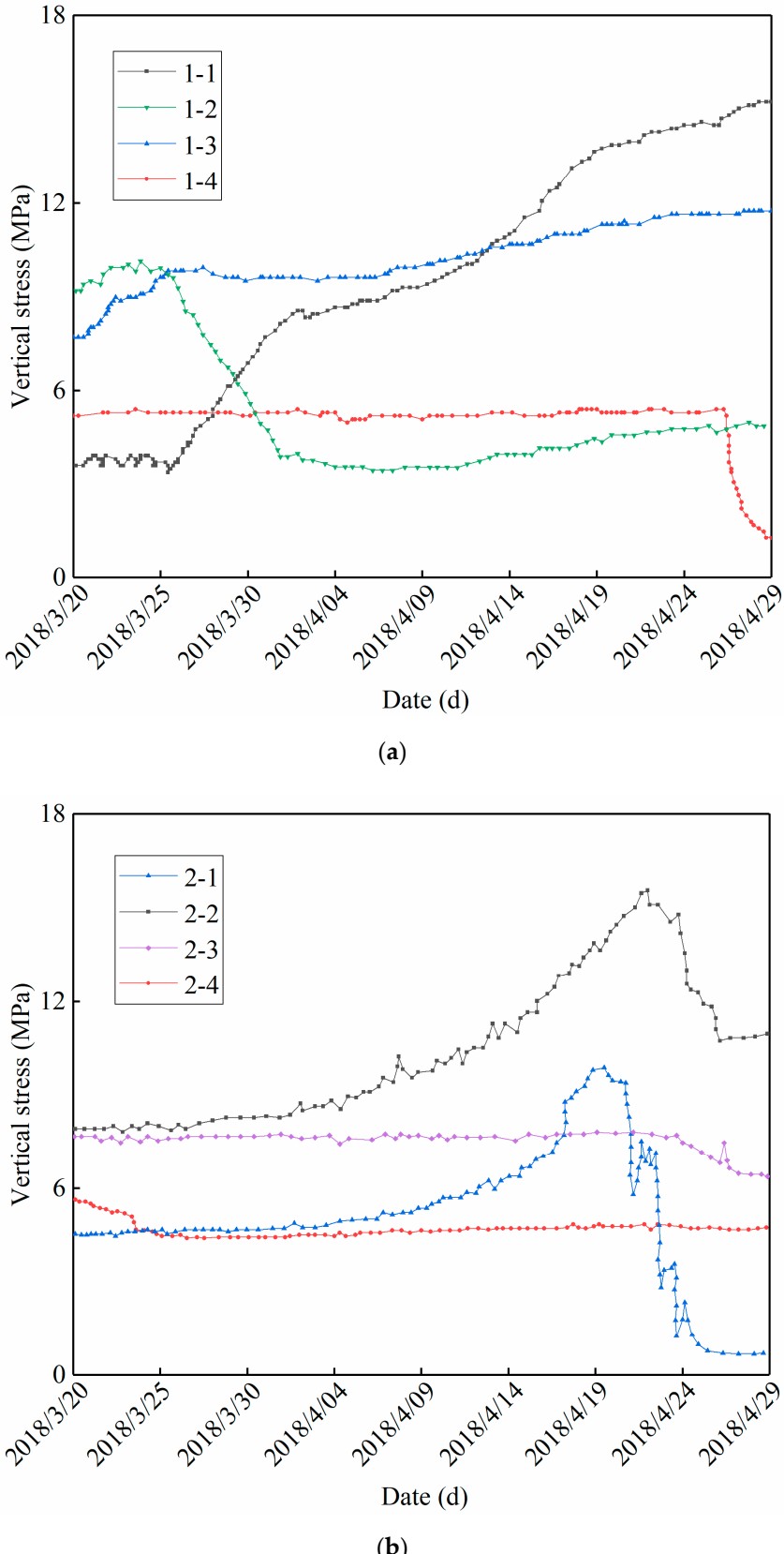

**Figure 18.** Relative vertical stress variation curve. (**a**) Relative vertical stress variation curve of station 1; (**b**) Relative vertical stress variation curve of station 2.

　　　Figure 18a showed that, the vertical stress of the 1-2 measuring point reached its peak on 3/23, when the working face was 20 m ahead of the measuring point. The vertical stress of 1-3 measuring point reached the first peak on 3/26, when the working face position was 3.7 m behind the measuring point, and began to decrease slightly on 4/2, when the working face position was 40 m behind the measuring point. With the advancement of the working face, the vertical stress of each measuring point increased periodically. On 4/27, the vertical stresses of measuring points 1-1, 1-2 and 1-3 tended to be stable, and the vertical stress of measuring point 1-4 dropped sharply. At this time, the working face was located 60 m behind station 1.

　　　Figure 18b showed that, the vertical stress of the measuring points 2-1, 2-2 and 2-3 began to rise on 3/26, when the working face was 96.3 m in front of station 2, and the vertical stress of the measuring point 2-1 reached its peak on 4/19, when the working face was 19 m in front of the measuring point. The vertical stress of the measuring point 2-2 reached its peak on 4/22, when the working face was 3 m behind the measuring point. The vertical stress of measuring point 2-3 increased slightly from 4/17 to 4/23 and decreased considerably on 4/24, when the working face advanced to 15 m behind the measuring point. From 4/18 to 4/25, the vertical stress of the 2-4 measuring point increased slightly, then decreased slightly. At this time, the working face was located 18 m behind the measuring point.

### 6.2.3. Comparative Analysis of Abutment Pressure

　　　In order to verify the numerical simulation results, the field monitoring data were compared with simulation results. When the working face advanced to station 3, the abutment pressure in front of the working face was shown in Figure 19. The field monitoring results showed that the abutment pressure in front of the working face increased first and then decreased, reaching a peak at 9 m ahead of the working face, and the stress distribution tends to be stable as it was farther away from the working face. The numerical simulation results also showed the distribution law of first increasing and then decreasing, reaching the stress peak at 10 m ahead of the working face, and the distribution law of abutment pressure was consistent with the monitoring results.

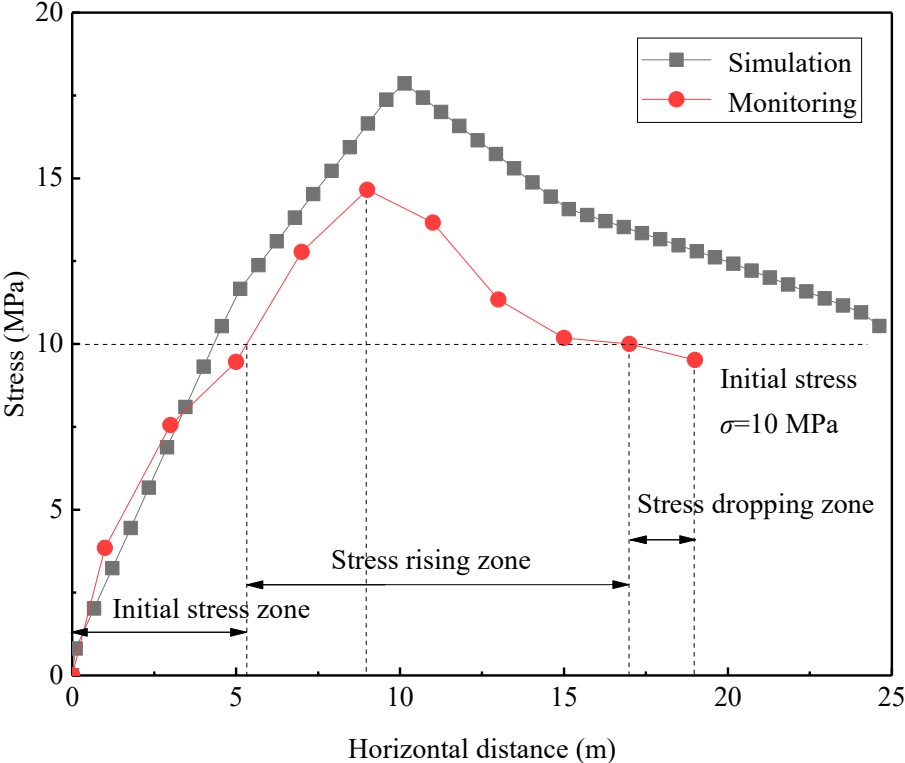

**Figure 19.** Distribution law of abutment pressure in front of the working face.

When the working face advanced to 146 m, station 2 lagged behind the working face by 28 m. The stress data extracted from the four measuring points could reflect the distribution characteristics of lateral abutment pressure, as shown in Figure 20. Field monitoring results showed that the lateral abutment pressure of the working face continued to increase within a range of 0–6 m from the roadway edge, reaching a peak value of 13.7 MPa at 6 m, and then gradually decreased. The numerical simulation results also showed a trend of first increasing and then decreasing, and reached the peak at 5 m away from the roadway edge. The stress distribution law was consistent with the monitoring results.

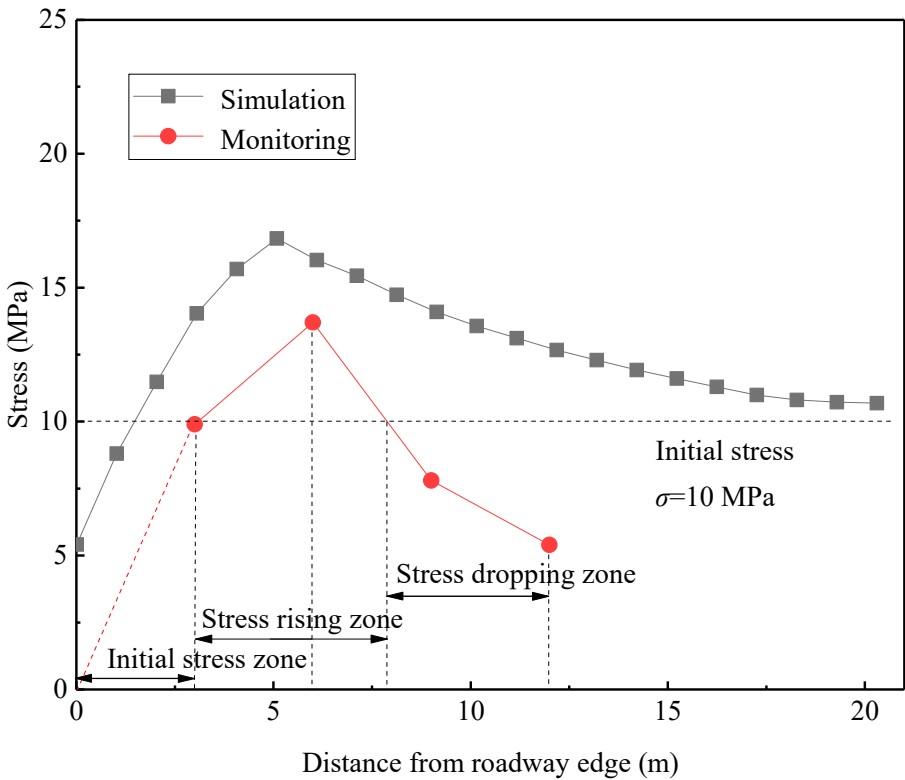

**Figure 20.** Distribution law of lateral abutment pressure in working face.

## 7. Conclusions

Taking Xiashanmao coal mine in Shanxi Province as the engineering background, the stress distribution in the process of coal seam mining was analyzed by establishing a numerical model, and the following conclusions were drawn:

(1) Based on the new technology of gob-side entry retaining by roof cutting without pillar, the mechanical model of the roof structure was established. Through the mechanical analysis of the model, the extension depth of the plastic zone on the solid coal side and the supporting load in the mining roadway were obtained under the condition of this technology, which provided certain theoretical support for the design of roadway support in the field.

(2) Through numerical simulation, the distributions of strike and lateral abutment pressure of the 9101 working face were obtained. Among them, the vertical stress distribution was as follows. In front of the working face: along the working face strike, the stress increased first, then decreased and then increased. The peak value of advance stress was formed at 10 m in front of the working face, and the peak value of non-pillar mining was reduced by 8–10% and 8–14% respectively compared with traditional mining. Along the inclination of the working face, the stress distribution of non-pillar mining and traditional mining was asymmetrical and symmetrical respectively, and the stress increased first and then decreased, while the stress at the side of roof cutting was significantly lower than that of the non-cutting. In the left side of the working face: along the inclination of the working face, the stress

of the non-pillar mining increased at first and then decreased, while the traditional mining showed the saddle-shaped distribution characteristics, and there were one and two peak stresses respectively. The peak stress of non-pillar mining was the smallest, which was 12–21% and 3–10% lower than that in the mining with large pillar and small pillar, respectively. The peak stress position in the former was closer to the mining roadway.

The horizontal stress distribution was as follows. In front of the working face: The stresses of the three increased first and then decreased. The peak position was 10 m ahead of the working face, in which the area, span and peak value of the stress rising area under large pillar mining were the largest, while those of non-pillar mining were the smallest. In the left side of the working face: high stress concentration was formed in the large pillar, not obvious in the small pillar, but a large area of stress concentration was formed in the 9102 working face. Because of the technology of roof cutting and retaining roadway, the phenomenon of stress increase appeared in the 9102 working face for non-pillar mining, but the area of the concentrated area was lower than that of traditional mining.

The results show that, in front of the working face: The stress increase area and peak stress of non-pillar mining were smaller than that of traditional mining. Around the retaining roadway, the stress transfer between roof rock beams was cut off by the roof cutting and pressure relief, which effectively weakens the stress concentration in deep surrounding rock. In the left side of the working face: The number of stress peaks was small and the peak stress was small for non-pillar mining. The stress-bearing areas of three mining methods were different, which were big coal pillar of large pillar mining, 9102 working face of small pillar mining and 9102 working face of non-pillar mining. The width of the plastic zone of surrounding rock of non-pillar mining was smaller and the bearing capacity was higher.

(3) The mine pressure monitoring data showed that the influence range of strike abutment pressure of the working face was 42.6–54.2 m. The distance between the peak position of abutment pressure and coal wall was 9–12 m. The sustained influence distance of abutment pressure was 33.6–42.2 m. The peak value of vertical stress at the deep-buried measuring point lagged behind that at the shallow-buried one. In the lateral side of the working face, the influence distance of mining in front of the 9101 working face was 48 m. With the advancement of the working face, the influence on it became more and more serious. The breaking and rotation of the hard strata overlying the working face caused the vertical stress of the shallow buried measuring point to rise. When 86 m ahead of the station, the stress at the measuring point was basically not affected by mining and reached a stable state. By comparing the field mine pressure monitoring results with the numerical simulation results, it could be found that the simulation results were consistent with the monitoring results.

**Author Contributions:** X.S. and Y.L. conceived and designed the research. Y.L. performed the numerical simulation. S.S. was involved in the construction of numerical model. J.W., X.C. and J.L. were involved in the numerical simulation. Y.L. analyzed the data and wrote the paper.

**Funding:** This work was supported by the National Key Research and Development Plan of China (2016YFC0600901), the National Natural Science Foundation of China (Grant No. 51874311), the Special Fund of Basic Research and Operating of China University of Mining & Technology, Beijing (Grant No. 2009QL03), which are gratefully acknowledged.

**Conflicts of Interest:** The authors declare no conflict of interest.

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
