# Peer review of "Study on Three-Dimensional Stress Field of Gob-Side Entry Retaining by Roof Cutting without Pillar under Near-Group Coal Seam Mining"

_processes, doi:10.3390/pr7090552_

Round 1

Reviewer 1 Report

Page 2 of 25

3rd paragraph

Line 3

It says:

Moreover, there is no need to remain pillars,

Should say:

Moreover, there is no need to leave pillars,

Page 3 of 25

2nd paragraph

Al quotes must be replaced without spacing before or after the words

It says:

In the 1960s and 1970s, longwall mining technology developed rapidly, and the " masonry beam

theory " was put forward, forming the " 121 " construction method of longwall mining (Qian 1981).

This technology requires two roadways to be tunneled for each working face, and large pillar is set

up to balance the underground pressure. The " transfer rock beam theory " was put forward by

analyzing the existence of internal and external stress field in high-stress area, forming the " 121 "

small pillar construction method of longwall mining (Song 1979; Zhu et al. 1982). However, the

traditional mining method of 121 will form hanging roof with insufficient collapse at the side of goaf,

and the roof subsidence and rotary deformation are large, which greatly affect the stability of pillar

and support system on roadway side. In order to reduce the development ratio, increase the coal

mining rate and improve the periodic pressure of the roof, the " cutting cantilever beam theory " was

born (He et al. 2015), and based on this theory, the mining technology of gob-side entry retaining by

roof cutting and pressure relief was proposed.

Should say:

In the 1960s and 1970s, longwall mining technology developed rapidly, and the "masonry beam

theory" was put forward, forming the "121" construction method of longwall mining (Qian 1981).

This technology requires two roadways to be tunneled for each working face, and large pillar is set

up to balance the underground pressure. The "transfer rock beam theory" was put forward by

analyzing the existence of internal and external stress field in high-stress area, forming the "121"

small pillar construction method of longwall mining (Song 1979; Zhu et al. 1982). However, the

traditional mining method of 121 will form hanging roof with insufficient collapse at the side of goaf,

and the roof subsidence and rotary deformation are large, which greatly affect the stability of pillar

and support system on roadway side. In order to reduce the development ratio, increase the coal

mining rate and improve the periodic pressure of the roof, the "cutting cantilever beam theory" was

born (He et al. 2015), and based on this theory, the mining technology of gob-side entry retaining by

roof cutting and pressure relief was proposed.

3rd paragraph, line 1

It says:

As mentioned above, the traditional mining method, namely " 121 " construction method of

Should say:

As mentioned above, the traditional mining method, namely "121" construction method of

Page 5 of 25

2nd paragraph, line 1

It says:

Explain the above picture as follows.

Should say:

The above picture is explained as follows.

4th paragraph, line 1

It says:

Where, x0:

Should say:

Where, x0:

Page 6 of 25

Figure 5a and Figure 5b

It says:

9101 haulage Roadway

Should say:

9101 haulage roadway

1st paragraph line 2

It says:

taiyuan formation

Should say:

Taiyuan Formation

Page 7 of 25

2nd paragraph, line 5

It says:

“ 121 “

Should say:

“121”

Figure 6

It says:

9101 haulage Roadway and 9102 haulage Roadway

Should say:

9101 haulage roadway and 9102 haulage roadway

Last line

It says:

The numerical calculation model was shown in Figure 7

Should say:

The numerical calculation model is shown in Figure 7

Page 8 of 25

Table 2

Columns 3 to 7 require width adjustment

Page 9 of 25

1st paragraph, line 2

It says:

mined step by step, and the goaf of 8 # coal seam

Should say:

mined step by step, and the goaf of 8 # coal seam

2nd paragraph, line 4

It says:

Figure 11

Shoud say: Figure 8

2nd paragraph, line 9

It says:

30 m, 80 m

Should say:

30 m, and 80 m

Page 10 of 25

Caption

It says:

Figure 11

Should say: Figure 8

Note: All subsequent figures require renumbering

Page 13 of 25

It says:

at 4, 5 and 6

Should say:

at 4, 5, and 6

It says:

in Figure. 15, 16 and 17, respectively.

Should say:

in Figures 12, 13, and 14, respectively.

Page 14 of 25

Graph (b)

It says:

Stress(MPa)

Horizontal  Distance(m)

Should say:

Stress (MPa)

Horizontal Distance (m)

Page 17 of 25

Graph (Figure 18 and Figure 19)

It says:

Figure 18 / Figure 19

Should say:

Figure 15 / Figure 16

It says:

Stress(MPa)

Distance from roadway edge(m)

Should say:

Stress (MPa)

Distance from roadway edge (m)

Page 18 of 25

Graph (Figure 20)

It says:

Figure 20

Should say:

Figure 17

It says:

stress(MPa)

Distance from roadway edge(m)

Should say:

Stress (MPa)

Distance from roadway edge (m)

1st paragraph

It says:

The vertical stress distribution curves at 10, 11, 12 and 13 measuring lines were shown in Figure

21, 22, 23 and 24, respectively.

Should say:

The vertical stress distribution curves at 10, 11, 12, and 13 measuring lines are shown in Figures

18, 192, 20, and 21, respectively.

2nd paragraph

Adjust Figure number sequence

Page 19 of 25 / 20 of 25 / 21 of 25 / 22 of 25

Graph (Figure 21, Figure 22, Figure 23, Figure 24)

It says:

Figure 21 / Figure 22 / Figure 23 / Figure 24

Should say:

Figure 18 / Figure 19 / Figure 20 / Figure 21

It says:

Stress(MPa)

Distance from roadway edge(m)

2nd paragraph (all)

Adjust Figure number sequence

Comments for the Authors:

1.       The results produced by the simulations are considering the economy of excavation and exclusion of pillar may provide acceptable pressure readings, however the function of a pillar in the middle is for physical safety whenever any unwanted extra pressure or a displacement of strata due to fault planes distinct from the direction of the seams. A lab or site test of the proposed model beyond the simulations is recommended for validating the proposed excavation approach.

2.       Safety procedures should be revised and tested (after a lab model or including more variables in the simulation.)

3.       Models with certain level of dip angle should be introduced in a future study and take note on the effects in the levels of stress in different directions.

4.       How to be sure that this approach is not prone to produce rockburst in the coal seams? What will be the safety procedures and recommended reinforcement given the new excavation pattern? How changes in the mining protective layer

5.       Results of Section 5.1.1 (Stress Distribution in From of Working Face) are presented in a repetitive way. Suggested to present a matrix-type chart with resulting values and illustrations, so they can be visualized and compared in a single view.

Author Response

Point 1: The results produced by the simulations are considering the economy of excavation and exclusion of pillar may provide acceptable pressure readings, however the function of a pillar in the middle is for physical safety whenever any unwanted extra pressure or a displacement of strata due to fault planes distinct from the direction of the seams. A lab or site test of the proposed model beyond the simulations is recommended for validating the proposed excavation approach.

Response 1: According to the review opinions, the field mine pressure monitoring is added in the paper for verification, please refer to section 6 for details.

Point 2: Safety procedures should be revised and tested (after a lab model or including more variables in the simulation).

Response 2: The effect of a series of safety procedures, such as pre-cracking roof cutting by energy-gathered blasting, constant resistance support and gangue support, are verified experimentally through setting up the test section on the engineering site. In the later stage, the effect of roadway retention and the stress and deformation of surrounding rock of roadway will be monitored in real time to verify the field effect.

Point 3: Models with certain level of dip angle should be introduced in a future study and take note on the effects in the levels of stress in different directions.

Response 3: In this paper, the numerical modeling was carried out based on the actual geological and structural conditions in the engineering site. The distribution characteristics of three-dimensional stress field of gob-side entry retaining by roof cutting without pillar were studied through numerical simulation, without considering the dip angle. However, the study of different dip strata and their effects on stress levels in different directions is indeed a direction worthy of further study. In the subsequent research work, relevant researches on the stress distribution characteristics of stope under the conditions of different dip angle models will be carried out, especially the researches on stress levels in different directions.

Point 4: How to be sure that this approach is not prone to produce rockburst in the coal seams? What will be the safety procedures and recommended reinforcement given the new excavation pattern? How changes in the mining protective layer

Response 4:

(1) In this paper, the stress evolution law of stope and surrounding rock of roadway of gob-side entry retaining by roof cutting without pillar was mainly studied, while the related research of rock burst was not carried out. However, there will be corresponding measures to deal with the problem of rock burst in the engineering site. The new mining system has perfect matching anti-impact measures, such as drillhole pressure relief, loose blasting, high-pressure water injection and so on. It can effectively deal with the problem of rock burst disaster and ensure the safety of coal mining.

(2) The high prestress constant-resistance-large-deformation anchor cable with structural characteristics of negative Poisson's ratio is a core technology of gob-side entry retaining by roof cutting without pillar. The new type anchor cable has the characteristics of high constant resistance, high prestress, large extension and high support strength, which has obvious advantages over the traditional anchor cable in dynamic pressure impact and large deformation control. Relying on the constant resistance device, the anchor cable can not only restrain the deformation through the high constant resistance value, but also absorb the deformation energy of the surrounding rock through the extension, realize the mechanical characteristics of the rigid-flexible coupling support of the anti-impact and constant resistance, and effectively control the engineering disasters such as rock burst (He et al., 2014, 2017).

(3) The reinforcement arrangement of constant-resistance-large-deformation anchor cables is carried out on the roof of the roadway, and the single hydraulic props in the roadway are also arranged in an encrypted arrangement.

1.        He, M.C.; Gong, W.L.; Wang, J.; Q, P.; Tao, Z.G.; Du, S.; Peng, Y.Y. Development of a novel energy-absorbing bolt with extraordinarily large elongation and constant resistance. International Journal of Rock Mechanics and Mining Sciences, 2014, 67, 29-42.

2.        He, M.C.; Li, C.; Gong, W.L.; Sousa, L.R.; Li, S.L. Dynamic tests for a Constant-Resistance-Large-Deformation bolt using a modified SHTB system. Tunnelling and Underground Space Technology, 2017, 64, 103-116.

Point 5: Results of Section 5.1.1 (Stress Distribution in From of Working Face) are presented in a repetitive way. Suggested to present a matrix-type chart with resulting values and illustrations, so they can be visualized and compared in a single view.

Response 5: According to the review opinions, the corresponding position has been modified in the paper. The revision can be found in Table 3, Table 4, Table 5, and Table 6 of Section 5.1 in the revised version.

Reviewer 2 Report

The topic of this article is interesting and important. It provides some findings but not enough needed for a publishable paper. The detailed comments are follows:

Section 2 and 3 are overly verbose and should be significant shortened. There are many rudimental knowledge and background that potential readers are already familiar with. Just present key statement and comparison between the present and conventional methods.

In Section 2.2, clearly define the current problem with parameters. Also, state physical meaning of Eqs. (1)-(5) and how to relate these formulation and results.

In numerical modeling, 1) clearly present the geometry of modeling. Which ground geometry to correspond to that of Fig. 7? 2) What is boundary conditions? 3) The current simulation scheme is confused. The construction sequence should be described in detail, using numbering.

The investigated location and stress field should be described in modelling part. The term “measuring line” may be incorrect because this study is not physical modelling, but numerical modelling.

Overall, the results and discussion is neither sufficiently solid nor well organized. Limited results are contained whereas lack of thorough analysis. It is difficult to understand what the authors wants to convey.

Author Response

Point 1: Section 2 and 3 are overly verbose and should be significant shortened. There are many rudimental knowledge and background that potential readers are already familiar with. Just present key statement and comparison between the present and conventional methods.

Response 1: In the revised version, sections 2 and 3 are simplified and the three parts of Section 2 are merged into two parts. For detailed modification, see the revised version.

Point 2: In Section 2.2, clearly define the current problem with parameters. Also, state physical meaning of Eqs. (1)-(5) and how to relate these formulation and results.

Response 2: For the comments put forward, the corresponding position has been modified in the revised version. The revision can be found in Section 2.1 of the revised version.

Point 3: In numerical modeling, 1) clearly present the geometry of modeling. Which ground geometry to correspond to that of Fig. 7? 2) What is boundary conditions? 3) The current simulation scheme is confused. The construction sequence should be described in detail, using numbering.

Response 3: According to the reviewing opinions, corresponding modifications have been made in the paper.

(1) The geometry of the model is shown in Figure 7 of Section 4.3.

(2) The boundary conditions of the model are shown in Figure 8 of Section 4.3.

(3) The detailed construction sequence is shown in Section 4.3.

Point 4: The investigated location and stress field should be described in modelling part. The term “measuring line” may be incorrect because this study is not physical modelling, but numerical modelling.

Response 4: Referring to the review opinions, the investigated position and stress field are described in Section 5.1, and the term “measuring line” is changed into the “monitoring line”.

Point 5: Overall, the results and discussion is neither sufficiently solid nor well organized. Limited results are contained whereas lack of thorough analysis. It is difficult to understand what the authors wants to convey.

Response 5: According to the review opinions, the conclusions have been condensed and summarized, and the conclusions have been revised to three parts, please refer to section 7 for details.

Reviewer 3 Report

The manuscript presents the study on stress field distribution of gob-side entry retaining by roof cutting and traditional remaining pillar based on theoretical analysis and numerical simulation. However, more considerations about the mechanical model of surrounding rock are needed. Accordingly, the authors are required to revise the manuscript. The following comments are requested to be addressed.

Major comments

1) Page 4 to 6: consider arching effect on the surrounding rock (Lee et al. 2006). 

Lee, C. J., Wu, B. R., Chen, H. T., & Chiang, K. H. (2006). Tunnel stability and arching effects during tunneling in soft clayey soil. Tunnelling and Underground Space Technology21(2), 119-132.

Minor comments

1)      A number of paragraphs start with prepositional phrase, e.g. As shown in Figure x. Please rephrase them. 

2)      Figures 5 and 6: need to improve the resolution of the figures.

3)      Figure 7: add indicators in the figure to describe each layer.

4)      Figure 11: avoid the overlapped dimension lines and separate them from the figure.

Author Response

Point 1: Page 4 to 6: consider arching effect on the surrounding rock (Lee et al. 2006).

Lee, C. J., Wu, B. R., Chen, H. T., & Chiang, K. H. (2006). Tunnel stability and arching effects during tunneling in soft clayey soil. Tunnelling and Underground Space Technology, 21(2), 119-132.

Response 1: According to the review opinions, the research on the mechanism of arch effect and its boundary conditions was studied, and the calculation of the plastic zone was referenced. Please refer to section 2.1 for details.

Point 2: A number of paragraphs start with prepositional phrase, e.g. As shown in Figure x. Please rephrase them.

Response 2: For the comments put forward, relevant modifications have been made in the paper.

Point 3: Figures 5 and 6: need to improve the resolution of the figures.

Response 3: Figs 5 and 6 are updated in the paper and changed to Figures 4 and 5 in the revised version.

Point 4: Figure 7: add indicators in the figure to describe each layer.

Response 4: According to the review opinions, the indicators are added in Figure 7 to describe each layer, as detailed in Figure 6 of the revised version.

Point 5: Figure 11: avoid the overlapped dimension lines and separate them from the figure.

Response 5: According to the review opinions, the dimension lines have been separated from the figure, please refer to Figure 9 of the revised version for details.

Round 2

Reviewer 2 Report

The revised manuscript was confirmed. Some comments are reflected but most are not.

The way they have presented their work is rather in reporting style than a scientifically discussed content. For example, the figures 10-23 seems to be similar while the monitoring lines are different. The quality of figures is not in professional level (e.g. Figure.8, newly added in the revision). The paper has several errors in engineering practice: incorrect SI unit expression, detailed statement on monitoring system (devices, number, and location) is missed. The comparison of numerical results and monitoring measurement is not given for model validation.

Author Response

Point 1: The figures 10-23 seems to be similar while the monitoring lines are different.

Response 1: For the comments put forward, Figures 10-23 in section 5.1 is adjusted, and pictures with similar stress distribution laws are deleted. The key parameters of stress distribution law are listed in tables, and the analysis of simulation results is simplified and refined. Please refer to section 5.1 of the revised version for details.

Point 2: The quality of figures is not in professional level (e.g. Figure.8, newly added in the revision).

Response 2: According to the reviewing opinions, Figure 8 has been modified, as detailed in Figure 8 of section 4.3 of the revised version.

Point 3: The paper has several errors in engineering practice: incorrect SI unit expression, detailed statement on monitoring system (devices, number, and location) is missed.

Response 3: According to the reviewing opinions, specific international units are added to the dimension markings in Figure 7 of section 4.3 and Figure 9 of section 5.1 of the revised version. The devices of monitoring system are introduced in detail, and the schematic diagram of monitoring system is added correspondingly (as shown in Figure 15 of section 6.1.1). The number and location of  monitoring system are described in detail in section 6.1.2 of the revised version, and the location of the monitoring device is shown in Figure 16.

Point 4: The comparison of numerical results and monitoring measurement is not given for model validation.

Response 4: According to the comments put forward, the field mine pressure monitoring results are compared with the numerical simulation results, and the results of numerical analysis are verified. Please refer to section 6.2.3 of the revised version for details.

Round 3

Reviewer 2 Report

The 2nd revision is properly reflected by the comments suggested by the reviewer.

Author Response

According to the previous two opinions, the article was reorganized.